# Quiescent neural stem cells transiently become neuron-like to coordinate long-range reactivation

Laura-Yvonne Gherghina 📍[1,2,3,6], Jocelyn L Y Tang[1,2,3,6], Leo Otsuki[2,3,4], Leia Judge[2,3,5] &
Andrea H Brand 📍[1,2,3] ✉

## Abstract

**Reactivation of quiescent neural stem cells (NSCs) in the central nervous system (CNS) is a tightly controlled process that generates new neurons and glia to maintain homeostasis or enable repair post-injury, but it remains unclear if reactivation of distinct NSC populations is coupled. Here, we discovered that NSC quiescence exit in *Drosophila* follows a hierarchical sequence, whereby activation of anterior stem cells in the brain lobes precedes and is required for the timely state-transition of more posterior NSCs in the ventral nerve cord. To achieve this, quiescent NSCs transiently activate neuronal genes. This transient neuronal state is temporary and specific to NSC dormancy, as neuronal genes are switched off after stem cells resume proliferation. Blocking neuronal firing in brain lobe neurons delays the onset of posterior NSC reactivation. Our results reveal long-range communication between quiescent NSCs to coordinate reactivation across the CNS, enabled by a transient, plastic neuron-like state that allows direct interaction with neuronal axons.**

**Subject Categories** Neuroscience; Stem Cells & Regenerative Medicine

## Introduction

Quiescence is an actively maintained state of reversible cell cycle arrest: cells stop proliferating and remain stalled in either the "$G_0$" or $G_2$ phase of the cell cycle (Otsuki and Brand, 2018; Buzgariu et al, 2014; Nguyen et al, 2017; Urbán et al, 2019). Both developing and adult tissues maintain pools of quiescent stem cells (de Morree and Rando, 2023). Cell cycle re-entry (reactivation) of quiescent stem cells occurs during growth and in response to physiological stimuli, to generate, or replace, differentiated cells. The ability of stem cells to generate new progeny raises the prospect of manipulating stem cells in vivo for brain repair or regeneration. To achieve this aim, however, requires knowledge how qNSCs respond to reactivation cues, which may be differ between spatial niches and across developmental time (Chaker et al, 2024).

Quiescent neural stem cells (NSCs) have been shown to reactivate in response to various signals, such as insulin signaling (Chell and Brand, 2010; Sousa-Nunes et al, 2011) or neuronal stimulation (Dittmann et al, 2025). However, little is known about how the reactivation state of individual NSCs may influence the behavior of other NSCs. In the adult mammalian brain, qNSCs lining the V-SVZ occupy distinct spatial niches that show differential responses to external stimuli (Chaker et al, 2024). However, it is not known whether qNSCs in different spatial niches cross-communicate with each other. Similarly, in the *Drosophila* central nervous system (CNS), NSCs are located in three major spatial niches along the anterior-to-posterior axis: brain lobes, thoracic ventral nerve cord (VNC), and abdominal VNC. Upon feeding, NSCs reactivate in response to glia-derived insulin signaling across the CNS (Chell and Brand, 2010; Sousa-Nunes et al, 2011). However, it is not known whether the subsequent exit from dormancy is intrinsic to each NSC or whether non-cell-autonomous signals exist between spatially defined NSC populations in the CNS.

NSCs are distributed across the CNS, raising the question of how NSC reactivation is coordinated temporally and spatially. NSCs undergo a dramatic morphological change during quiescence, extending projections into the axon-dense neuropil (Truman and Bate, 1988; Chell and Brand, 2010). Upon reactivation, the NSCs retract these projections and resume asymmetric cell division to produce neurons and glia (Truman and Bate, 1988; Tsuji et al, 2008). The qNSC projection is extended toward the neuropil, raising the possibility that qNSCs might communicate with neurons. Neuronal activity is known to influence NSC proliferation in the mammalian NSC niches as changes in membrane potential, neuronal electrical activity, or neurotransmitter release can modulate the proliferation of neural progenitors (Dittmann et al, 2025). However, the relationship between neuronal signaling and the spatiotemporal coordination of NSC reactivation is unknown.

Here, we show that reactivation of qNSCs relies on communication with neurons, which mediate the relay of reactivation in a temporally and spatially defined sequence. This enables NSCs to

[1]Department of Cell Biology and Regenerative Medicine Institute, NYU Grossman School of Medicine, New York, NY, USA. [2]Gurdon Institute, University of Cambridge, Cambridge, UK. [3]Department of Physiology, Development and Neuroscience, University of Cambridge, Cambridge, UK. [4]Present address: Hubrecht Institute – KNAW and University Medical Center Utrecht, Utrecht, The Netherlands. [5]Present address: Springer Nature, 4 Crinan Street, London, UK. [6]These authors contributed equally: Laura-Yvonne Gherghina, Jocelyn L Y Tang. ✉E-mail: andrea.brand@nyulangone.org

coordinate their reactivation across the entire central nervous system, from the brain lobes to the nerve cord. We found that, whilst maintaining the expression of stem cell genes, qNSCs adopt a neuronal morphology and express a host of neuronal genes. We show that the synaptic protein Hig is present along the NSC projection and is required for reactivation. Our results uncover qNSC plasticity that allows for neuronally mediated coordination of timely reactivation between spatial niches.

# Results

## NSCs reactivation occurs in an anterior-to-posterior sequence

During *Drosophila* development, neural stem cells enter quiescence in late embryogenesis and reactivate in early larval development in response to nutrition, giving rise to neurons and glia that contribute to the adult nervous system. We previously found that 75% of NSCs arrest in the $G_2$-phase of the cell cycle, and the remaining 25% arrest in $G_0$ (Otsuki and Brand, 2018). We showed that heterogeneity in NSC cell cycle arrest leads to differences in reactivation timing, with $G_2$-arrested cells reactivating much earlier than their $G_0$-arrested counterparts (Otsuki and Brand, 2018).

In early larval development, feeding induces the secretion of insulin-like peptides from a glial niche, which leads to NSC reactivation from quiescence (Chell and Brand, 2010; Sousa-Nunes et al, 2011). qNSC reactivation initiates in response to feeding, beginning with cell growth (Fig. 1A). We sought to understand how qNSCs coordinate their timing of reactivation. We profiled the distribution of $G_2$ and $G_0$-arrested cells in the brain lobes and the VNC. We found that the 3:1 ratio of $G_2/G_0$ arrested qNSCs is maintained in the brain lobes and the thoracic VNC, whereas the abdominal NSCs are equally split between $G_2$ and $G_0$ arrest (Fig. 1B). We assessed reactivation based on expression of the protein Worniu (Wor). While the *wor* transcript is expressed in NSCs throughout development, Wor protein is only present in NSCs upon reactivation, prior to mitosis (Otsuki and Brand, 2018). We found that qNSCs reactivate first in the brain lobes (4 h after larval hatching—ALH), followed by qNSCs in the thoracic (8 h ALH) and abdominal (20 h ALH) VNC (Fig. 1C,D). We showed previously that $G_2$-arrested cells reactivate before $G_0$-arrested cells (Otsuki and Brand, 2018), but this does not account for the anterior-to-posterior sequence that begins in the brain lobes (Fig. 1D). Intriguingly, qNSCs in the abdominal VNC remain quiescent more than fifteen hours longer than qNSCs in the brain lobes, despite the secretion of insulin-like peptides across the entire CNS emanating from the glial-niche (Chell and Brand, 2010). This suggests that there must be an additional mechanism regulating reactivation.

To achieve this sequence of reactivation, qNSCs may either be able to determine their specific reactivation timings cell-autonomously, or the process could be coordinated via non-cell-autonomous signals from the brain to the VNC.

## NSCs propagate reactivation along the anterior-to-posterior axis

Do qNSCs determine their reactivation timing cell-autonomously, or does the process rely on long-range signals from the brain lobes

to the VNC? To answer this question, we generated a genetic tool to restrict GAL4-mediated gene expression to NSCs in the brain lobes (Fig. 2A), by combining the NSC-specific driver *wor*-GAL4 with *tsh*-GAL80, which prevents GAL4 expression in the VNC (Simpson, 2016).

First, we tested first whether inhibiting the insulin signaling pathway in the brain lobes would impair reactivation of qNSCs in the ventral nerve cord. Upon misexpression of an inhibitor of insulin signaling (PTEN) in the brain lobe qNSCs, we found that reactivation of ventral nerve cord qNSCs was severely impaired (Fig. 2B).

Next, we cultured animals under "starvation" conditions in which qNSCs do not reactivate (Chell and Brand, 2010). We reactivated qNSCs exclusively in the brain lobe qNSCs by expression of constitutively active AKT, thereby activating the insulin signaling pathway. Remarkably, this was sufficient to induce reactivation of the ventral nerve cord qNSCs (Fig. 2C). Therefore, to coordinate reactivation between the brain lobes and the VNC, qNSCs appear to be able to propagate a signal along the anterior–posterior axis of the CNS.

## Hyperpolarization and depolarization of qNSCs also modulate reactivation

In the mammalian brain, it has been shown that hyperpolarization of neural progenitors and NSCs can alter neurogenesis (Vitali et al, 2018) and proliferation (Wang et al, 2008). These observations led us to ask whether changes in bioelectric properties of qNSCs would influence proliferation and whether these changes would impact the reactivation of qNSCs in the VNC. To assess this, we overexpressed Kir2.1, an inward-rectifying potassium channel that lowers resting membrane potential (Urrego et al, 2014). We found that misexpression of Kir2.1 in the brain lobes (Fig. 3A) led to impaired reactivation of VNC qNSCs (Fig. 3B). Thus, hyperpolarization of qNSCs in the brain lobes is sufficient to impair reactivation of qNSCs in the VNC.

Since electrical inhibition of qNSCs prevented reactivation, we tested whether depolarization of brain lobe NSCs would enhance cell cycle re-entry in the VNC. We expressed the temperature-activated $Ca^{2+}$ channel dTrpA1 (Pulver et al, 2009) in brain lobe qNSCs and found increased reactivation in the VNC based and found increased reactivation as shown by pH3 labeling and expression of Worniu (Fig. 3C). Therefore, the reactivation of brain lobe NSCs is relayed to NSCs in the VNC and promotes their reactivation. This suggests that while depolarization of the anterior brain lobe qNSCs may accelerate reactivation, perhaps working in concert with other processes. These data indicate that altering the bioelectric properties of brain lobe NSCs also impacts reactivation of NSCs in the VNC.

## Quiescent neural stem cells take on the properties of neurons

At the stage at which qNSCs enter quiescence, the cells are similar in size to neurons, and their projections resemble axons (Fig. 4A). In addition, we showed that hyperpolarization and depolarization of qNSCs impact reactivation. We performed single-cell RNA-sequencing to identify quiescence-specific genes that might allow more insight into how qNSCs communicate with one another (Figs. 4B and EV1).

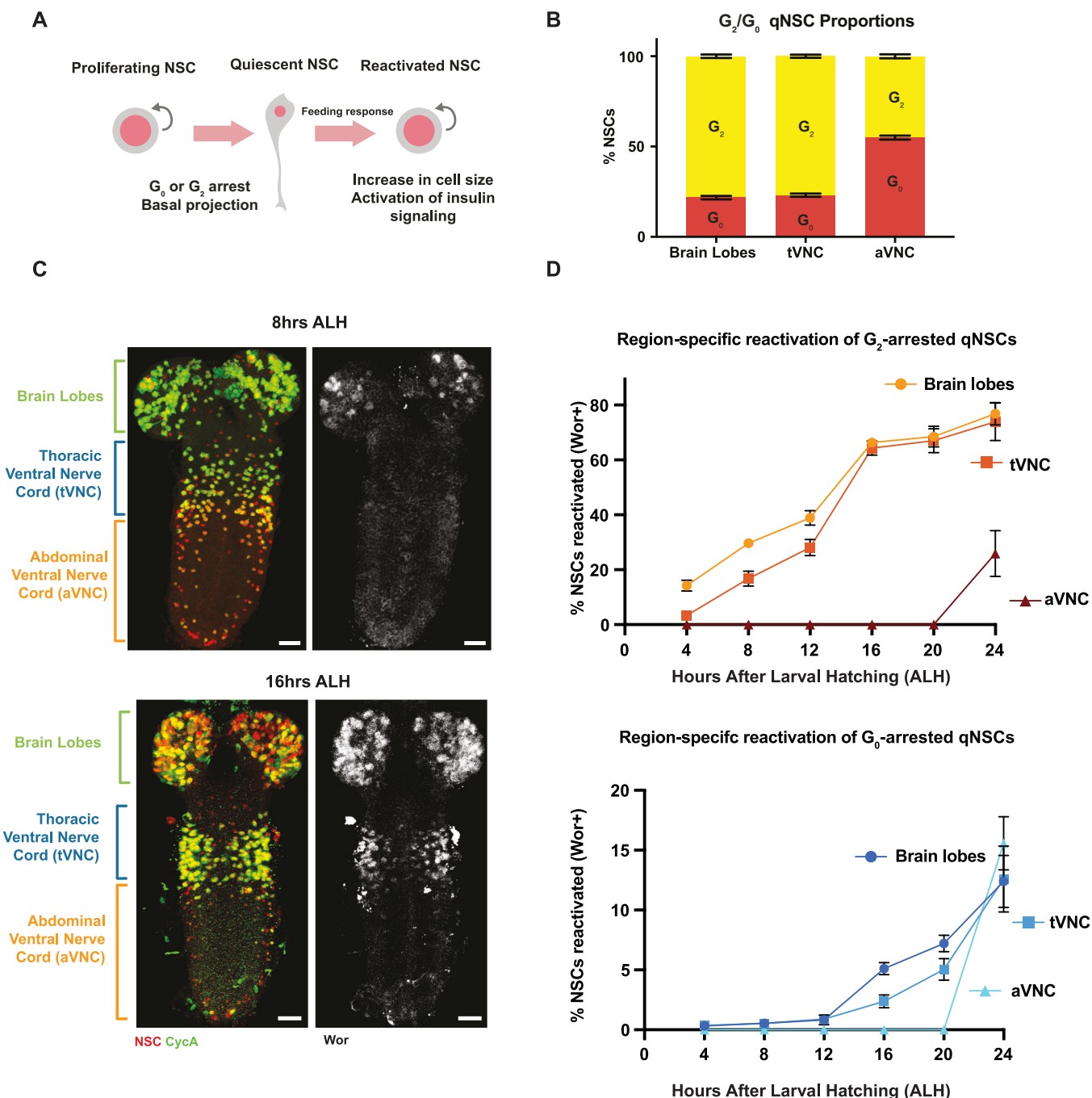

**Figure 1. Quiescent NSCs reactivate in an anterior-to-posterior sequence.**

(A) Illustration of characteristics specific to quiescent and reactivated NSCs. (B) Proportion of NSCs arrested in $G_2$ or $G_0$ in the brain lobes, thoracic VNC (tVNC), and abdominal VNC (aVNC). $n = 10$ brains; error bars indicate SD, and the center is the mean. (C) Expression of the reactivation indicator Worniu (Wor) in the brain lobes at 8 h after larval hatching (ALH) followed by tVNC expression at 16 h ALH; Worniu protein is only expressed in proliferating neural stem cells (Otsuki and Brand, 2018), as an early indicator of reactivation; scale bars, 20 μm; Deadpan (Dpn, NSC marker) in red, Cyclin A (CycA) in green, Wor in white. (D) Percentage of NSCs reactivated (Wor +) between 0 h and 24 h ALH assayed at 4-h intervals, split by re-entry from $G_2$ or $G_0$ arrest; $n = 4$-10 brains, error bars indicate SEM the center is the mean. Source data are available online for this figure.

As expected, both quiescent and reactivated NSCs expressed neural stem cell genes such as *deadpan*, *worniu*, and *klumpfuss*. Surprisingly, we found that qNSCs also express genes characteristic of neurons (Figs. 4C–E and EV2A,B). Neuronal gene expression

was only observed in qNSCs; upon reactivation, NSCs reverted to stem cell gene expression, and neuronal genes were silenced.

GO term analysis of qNSC gene expression revealed an enrichment for neuronal genes involved in neurotransmitter release, synaptic

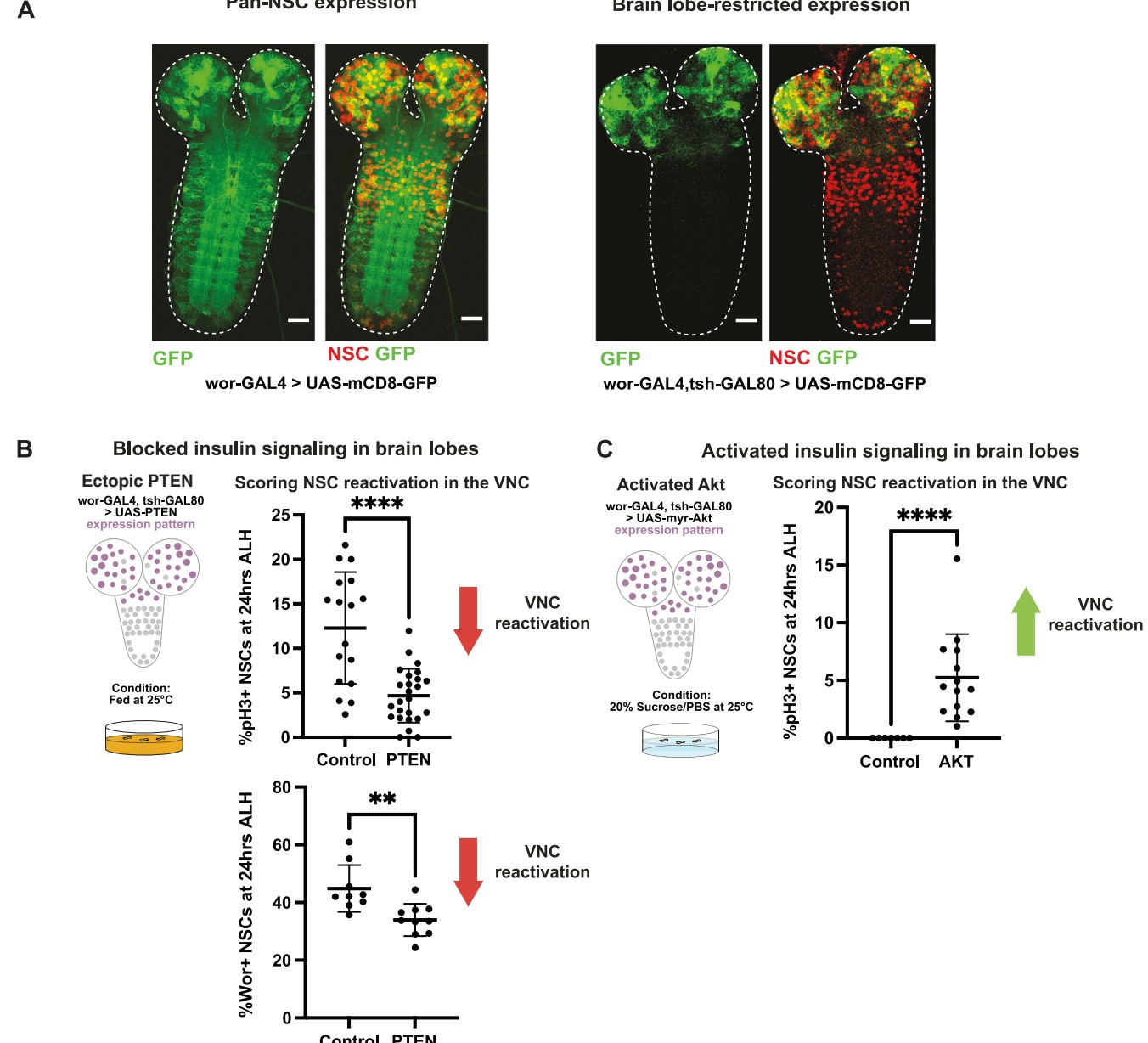

**Figure 2. NSC reactivation is non-cell-autonomous.**

(A) Ectopic gene expression is restricted to the brain lobes using *wor-GAL4, tsh-GAL80*; scale bar, 20 μm; mCD8-GFP in green, Dpn (NSC) in red, 24 h ALH. White dashed lines indicate the CNS outline. (B) PTEN misexpression in the brain lobes under fed conditions leads to decreased reactivation in the VNC as shown by pH3 labeling and Wor expression; $n = 17$ (control, pH3), $n = 25$ (PTEN, pH3), $n = 9$ (control, Wor), $n = 10$ (PTEN, Wor), P value = 0.0021. (C) Misexpression of constitutively active Akt under amino acid starvation conditions in the brain lobes is sufficient to induce reactivation in the VNC; $n = 7$ (control), $n = 14$ (Akt) Mann–Whitney U test; **P < 0.01, ****P < 0.0001. Error bars indicate SD; the center is the mean. Source data are available online for this figure.

assembly and synaptic activity (Fig. 4B). In contrast, reactivated NSCs expressed genes for transcription and translation (Fig. 4B). Quiescent, but not reactivated, neural stem cells expressed neuronal genes involved in electrochemical processes, including GABAergic (*Gad1, Rdl*), cholinergic (*nAChRalpha6, mAChR-A*) and glutamatergic neurotransmission (*VGlut*; Fig. 4C). This suggests that qNSCs are heterogenous and may communicate with different neuronal subtypes.

We generated a "neuron score" based on the top neuronal genes expressed in neurons during late embryogenesis and found that qNSCs

have a high neuron score, whereas reactivated NSCs do not exhibit neuronal gene expression (Fig. 4E). Therefore, qNSCs transiently become neuronal while maintaining expression of stem cell genes.

## The synaptic protein *Hikaru genki* (Hig) is required for reactivation

While there have been previous indications that quiescent neural stem cells may respond to secreted neurotransmitters (Bao et al, 2017; Song

## A    Brain lobe-specfic neural stem cell electrical inhibition

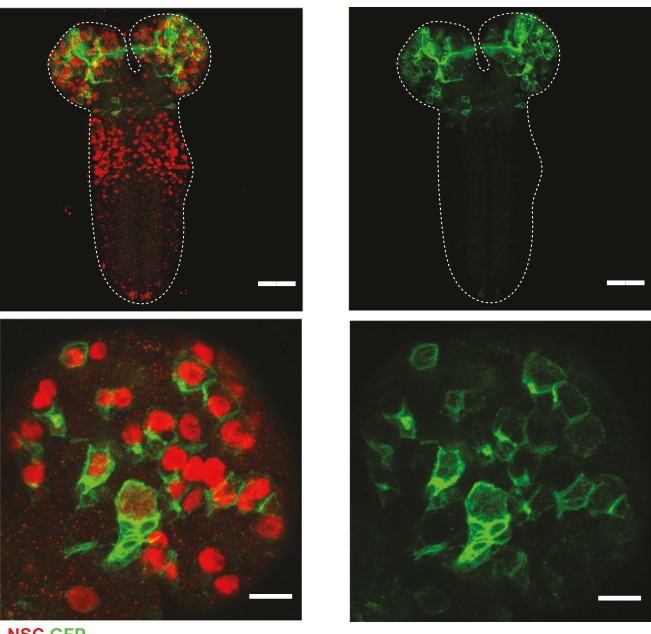

NSC GFP

**wor-GAL4, tsh-GAL80 > UAS-Kir2.1-GFP**

## B    Blocking electrical stimulation in brain lobe NSCs

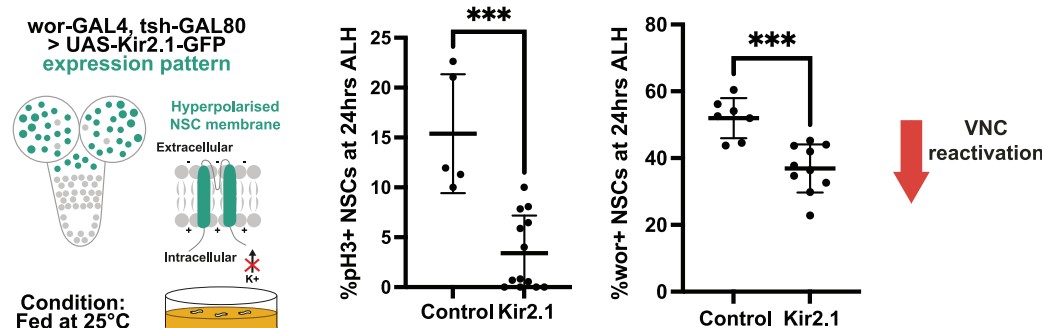

Scoring NSC reactivation in the VNC

## C    Promoting electrical stimulation in brain lobe NSCs

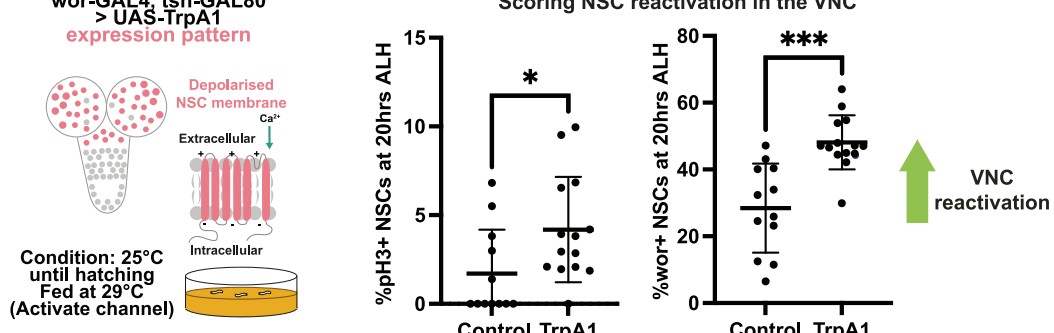

Scoring NSC reactivation in the VNC

◄ **Figure 3. Quiescent NSCs are electrically sensitive.**

(A) Misexpression of the potassium channel Kir2.1-GFP in the brain, scale bars, 40 µm; brain lobe expression of Kir2.1-GFP, scale bars, 10 µm, 24 h ALH. White dashed lines indicate the CNS outline. (B) Hyperpolarization of brain lobe NSCs leads to a delay in reactivation in the VNC; $n = 8$ (control, pH3), $n = 15$ (Kir2.1, pH3), $P$ value $= 0.0052$; $n = 10$ (control, Wor), $n = 12$ (Kir2.1, Wor). (C) Depolarization of brain lobe NSCs using the temperature-sensitive dTrpA1 channel speeds up reactivation as indicated by pH3 and expression of the reactivation marker Worniu, $n = 12$ (control pH3/Wor), $n = 14$ (TrpA1 pH3/Wor), $P$ value (pH3) $= 0.0207$, $P$ value (Wor) $= 0.0001$. Mann–Whitney $U$ test; $*P < 0.05$, $**P < 0.01$, $***P < 0.001$, $****P < 0.0001$. Error bars indicate SD; the center is the mean. Source data are available online for this figure.

et al, 2012), no synaptic mechanism for this interaction has been previously proposed (Dittmann et al, 2025). Given the mixed "stem cell-neuron" identity of qNSCs, we tested whether downregulation of a gene involved in synaptic communication was required for reactivation, which would suggest NSC-neuron direct communication.

Among the neuronal genes upregulated in qNSCs, we found several cholinergic receptors: *nAChRalpha6, nAChRalpha5, nAChRalpha1, mAChR-A* (Fig. 5A). In addition, we found that the cholinergic synapse-specific genes (Nakayama et al, 2014, 2016) *hikaru genki* (hig) is expressed in qNSCs and downregulated upon reactivation (Fig. 5A–C). In the adult *Drosophila* brain, Hig accumulates at cholinergic synapses and is required for the assembly of cholinergic receptors at the synapse (Nakayama et al, 2014). In qNSCs, Hig is found along the projection and in the neuropil (Fig. 5D). We found that knocking down *hig* in NSCs impairs reactivation (Fig. 5E). Downregulation of *hig* in neurons also significantly impairs qNSC reactivation (Figs. 5F and EV3). Moreover, we found that neuronal knockdown of the *hig anchoring protein, hasp*, which enables the segregation of Hig at cholinergic synapses (Nakayama et al, 2016) also impairs qNSC reactivation (Fig. EV4). Thus, the synaptic proteins Hig and Hasp are both required for timely NSC reactivation, suggesting that a synaptic mechanism may be involved.

## Neuronal activity is required for neural stem cell reactivation

As previously shown, qNSCs extend long cellular projections towards the pre-existing neuropil. These projections are retracted once the stem cells reactivate and resume proliferation (Tsuji et al, 2008). When fully extended, the termini of qNSC projections can be seen close to axonal tracts in the CNS (Fig. 6A,B). Given that qNSCs express neuronal genes, including the synaptic protein Hig, which we have shown to be required for reactivation, we investigated whether the ventral cord qNSCs receive signals directly from axons descending from the brain lobes.

To test whether neurons in the brain lobes mediate NSC to NSC communication, we specifically silenced brain lobe neurons, which send descending tracts into the VNC, by expressing the potassium channel Kir2.1 (Baines et al, 2001; Berni et al, 2012) (Fig. 6C). Blocking neuronal firing drastically delayed the onset of reactivation as marked by Worniu (Fig. 6D) and completely abolished the proliferation of qNSCs in the ventral nerve cord, which was assessed using pH3 staining (Fig. 6D). This demonstrates that neuronal activity is required for the non-autonomous reactivation of posterior qNSCs.

## Discussion

Here we show that, in addition to the intrinsic temporal heterogeneity that exists between the earlier-reactivating $G_2$-arrested and later-reactivating $G_0$-arrested cells (Otsuki and Brand, 2018), qNSCs exit dormancy in an anterior-to-posterior sequence. qNSCs coordinate their spatially and temporally defined sequence of reactivation non-cell-autonomously, beginning in the brain lobes (anteriorly) followed by the ventral nerve cord (posteriorly). We found that reactivation of the posterior stem cells is dependent upon prior reactivation of brain lobe stem cells, even in the presence of reactivation signals induced by feeding.

In the mouse adult brain, depending on their spatial localization, NSCs give rise to different neuronal and glial subtypes (Fiorelli et al, 2015; Chaker et al, 2024). Similarly, in *Drosophila*, the asynchronous reactivation of qNSCs could allow for the staggered formation of subsequent neuronal circuits (Otsuki and Brand, 2018). A crucial role in maintaining and reactivating neural stem cells is played by niche-derived cues (Urbán et al, 2019), with neuronal innervation emerging as an important regulator of NSC proliferation (Dittmann et al, 2025). It has been proposed that the manner in which adult NSCs respond to external stimuli, including neuronal innervation, may differ across spatial niches (Chaker et al, 2024). However, it is unclear whether NSCs may influence one another's exit from quiescence. We find that NSCs communicate across long distances: reactivating NSCs in the brain lobes relay signals to qNSCs in the ventral nerve cord via descending axonal tracts (Fig. 7A,B). These descending axons are contacted by qNSC cellular projections, and the relay of reactivation relies on the activity of these brain lobe neurons. We show that qNSCs express neuronal genes that would allow for communication with neurons, including via synapses. We find that two synaptic matrix genes, *hig* and *hasp*, are required for timely NSC reactivation. Interestingly, *hig* expression is required in both NSCs and neurons for timely stem cell reactivation. As in the adult brain (Nakayama et al, 2014, 2016), the secretion of the matrix proteins Hig and Hasp by qNSCs and neurons might enable the accumulation of cholinergic receptor subunits at qNSC-neuron junctions. Whether qNSCs and neurons transiently communicate via synapses or through other structures remains to be determined. We show that NSCs display remarkable cell fate plasticity, transiently taking on neuronal characteristics during the quiescence period whilst continuing to express neural stem cell genes. However, qNSCs appear to be heterogenous and the contacts between NSCs and neurons may be specific to different NSC and neuronal subpopulations. Future work will explore how distinct neuronal circuits regulate stem cell reactivation within the anterior-to-posterior sequence. Other mechanisms, such as direct NSC-NSC or NSC-glia communication, may also play a role in coordinating reactivation.

Neuron-NSC interactions may also be relevant for cancer progression and pathology. Glioma cells, which have quiescent subpopulations (Antonica et al, 2022; Ishii et al, 2016; Deleyrolle et al, 2011), have been observed to form synapses with neurons in the surrounding microenvironment (Venkataramani et al, 2019;

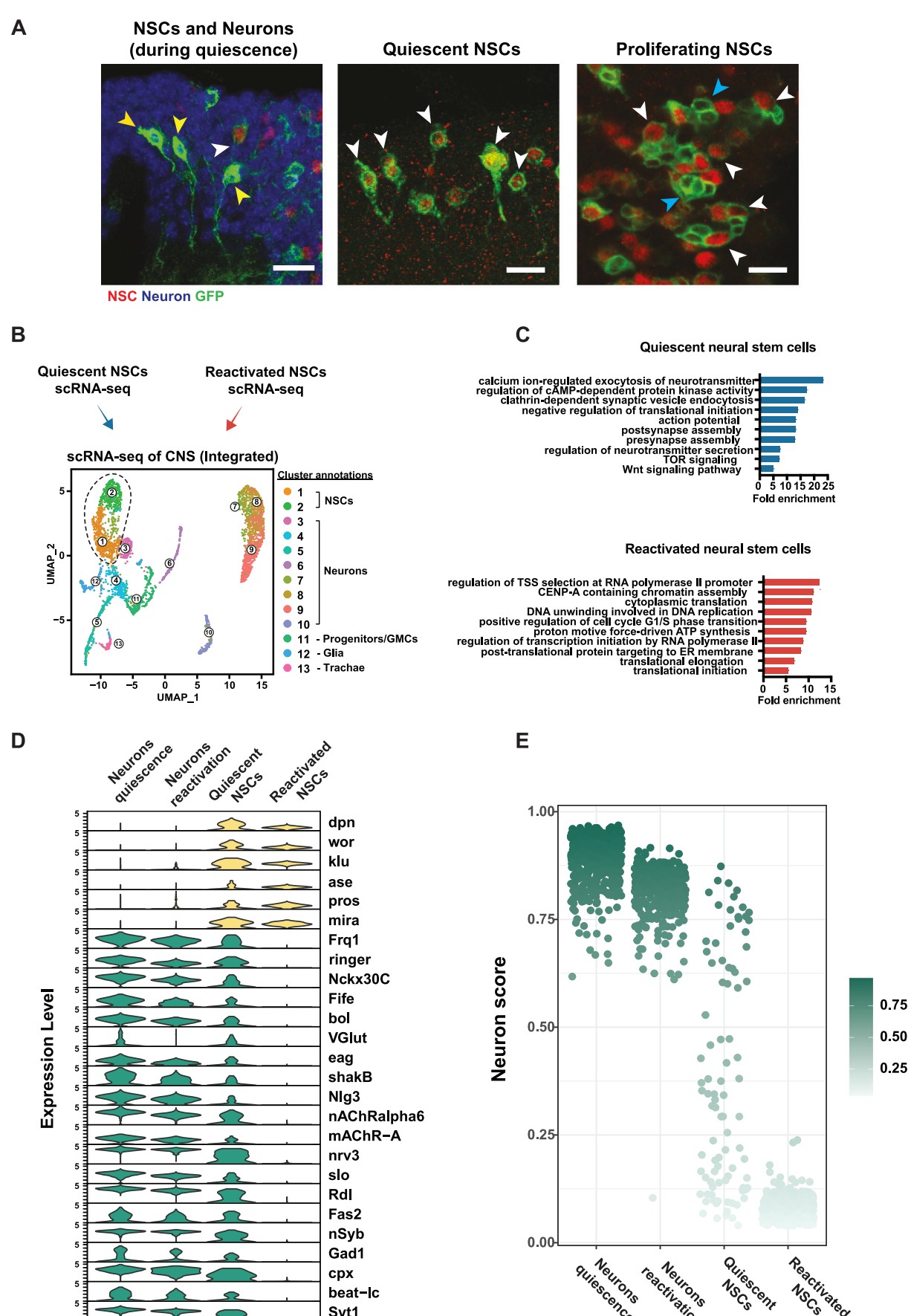

**A** NSCs and Neurons (during quiescence) | Quiescent NSCs | Proliferating NSCs

NSC Neuron GFP

**B** Quiescent NSCs scRNA-seq | Reactivated NSCs scRNA-seq

scRNA-seq of CNS (Integrated)

Cluster annotations
1 } NSCs
2
3
4
5
6
7 } Neurons
8
9
10
11 - Progenitors/GMCs
12 - Glia
13 - Trachae

**C** Quiescent neural stem cells

calcium ion-regulated exocytosis of neurotransmitter
regulation of cAMP-dependent protein kinase activity
clathrin-dependent synaptic vesicle endocytosis
negative regulation of translational initiation
action potential
postsynapse assembly
presynapse assembly
regulation of neurotransmitter secretion
TOR signaling
Wnt signaling pathway

Fold enrichment

Reactivated neural stem cells

regulation of TSS selection at RNA polymerase II promoter
CENP-A containing chromatin assembly
cytoplasmic translation
DNA unwinding involved in DNA replication
positive regulation of cell cycle G1/S phase transition
proton motive force-driven ATP synthesis
regulation of transcription initiation by RNA polymerase II
post-translational protein targeting to ER membrane
translational elongation
translational initiation

Fold enrichment

**D** Neurons quiescence | Neurons reactivation | Quiescent NSCs | Reactivated NSCs

dpn, wor, klu, ase, pros, mira, Frq1, ringer, Nckx30C, Fife, bol, VGlut, eag, shakB, Nlg3, nAChRalpha6, mAChR-A, nrv3, slo, Rdl, Fas2, nSyb, Gad1, cpx, beat-Ic, Syt1

Expression Level

**E** Neuron score

Neurons quiescence | Neurons reactivation | Quiescent NSCs | Reactivated NSCs

**Figure 4. Quiescent NSCs express neuronal genes.**

(A) Quiescent NSCs bear a projection that morphologically resembles neuronal axons, which is not present in proliferating NSCs. White arrows indicate NSCs, yellow arrows indicate neurons, and blue arrows indicate NSC progeny. NSCs are marked by Dpn in red, neurons are marked by Elav in blue and membranes are marked using *wor*-GAL4 > mCD8-GFP (left-hand side panels) showing quiescent aVNC NSCs and proliferative tVNC NSCs at 24 h ALH and *eg*-GAL4 > UAS-mCD8-GFP (rightmost panel) showing qNSCs and neurons at 0 h ALH. Scale bars, 10 μm. (B) scRNAseq during quiescence and after reactivation captures both NSC and neuronal populations. (C) GO terms for quiescent NSCs and reactivated NSCs. (D) Expression of NSC and neuronal genes in neurons during quiescence, neurons after reactivation, quiescent NSCs, and reactivated NSCs. (E) Neuron score obtained using UCell comparing neurons and NSCs during quiescence and reactivation. Source data are available online for this figure.

Venkatesh et al, 2019). Electrochemical signals have been shown to influence the proliferation of glioma cells, which form synapses with glutamatergic neurons (Venkataramani et al, 2019). In pediatric gliomas, depolarization of glioma tumor cells via optogenetics promotes proliferation (Taylor et al, 2023). In some cases, the formation of synapses between glioma cells and neurons also impacts the frequency of seizures, a common symptom in glioma patients (Yu et al, 2019). Here, we provide evidence of a physiological instance in which cell fate plasticity allows non-neuronal cells to become neuronal-like to communicate with neurons, which may be relevant for disease mechanisms.

Our study has identified a non-cell-autonomous spatial hierarchy by which quiescent NSCs coordinate timely reactivation. This mechanism is mediated via NSC-neuron communication, achieved through the transient acquisition of neuronal characteristics by quiescent NSCs. Further investigation of specific contacts will shed light on NSC-neuron communication during brain development, extending towards how these mechanisms may be involved in disease.

# Methods

### Reagents and tools table

| Reagent/resource | Reference or source | Identifier or catalog number |
|---|---|---|
| **Experimental models** | | |
| *Drosophila melanogaster:* | | |
| wor-GAL4 | Albertson et al, 2004 | |
| elav-GAL4 | Luo et al, 1994 | |
| grh-GAL4 | Chell and Brand, 2010 | |
| UAS-mCD8-GFP | Lee and Luo, 1999 | |
| tubGAL80ts | McGuire et al, 2003 | |
| tsh-GAL80 | Kind gift from Gero Miesenboeck | |
| UAS-myr-AKT | Stocker et al, 2002 | |
| UAS-PTEN | Huang et al, 1999 | |
| UAS-Kir2.1-GFP | Baines et al, 2001 | |
| UAS-TrpA1 | BDSC | 26263 |
| UAS-higRNAi | BDSC | 42000 |
| UAS-mCD8-mCherry | BDSC | 27391 |
| UAS-mCherryRNAi | BSDC | 35785 |

| Reagent/resource | Reference or source | Identifier or catalog number |
|---|---|---|
| UAS-Dcr2 | Dietzl et al, 2007 | |
| UAS-haspRNAi #1 | VDRC | v30767 |
| UAS-haspRNAi #2 | VDRC | v101250 |
| eg-GAL4 | Ito et al, 1995 | |
| **Antibodies** | | |
| Chicken anti-GFP | Abcam | ab13970 |
| Guinea pig anti-Dpn | Caygill and Brand, 2017 | |
| Rabbit anti-pH3 | Millipore | 06-570 |
| Rat anti-Wor | Abcam | ab195173 |
| Mouse anti-Fas2 | DSHB | 1D4 |
| Rabbit anti-CycA | Kind gift from David Glover (Whitfield et al, 1990) | rb270 |
| Rabbit anti-RFP | Abcam | ab62341 |
| Rat anti-Elav | DSHB | 7E8A10 |
| Guinea pig anti-Hig | Kind gift from Dr Chihiro Hama, Kyoto Sangyo University, Japan (Nakayama et al, 2014) | |
| Rat anti-Hasp | Kind gift from Dr Chihiro Hama, Kyoto Sangyo University, Japan (Nakayama et al, 2016) | |
| Rabbit anti-Rx | Kind gift from Uwe Walldorf. (Davis et al, 2003) | |
| Alexa Fluor 405, 488, 546, 568, 633 | Life Technologies | |
| DyLight 405 | Jackson Laboratories | |
| **Oligonucleotides and other sequence-based reagents** | | |
| *dpn* HCR probe | Molecular Instruments | |
| *hig* HCR probe | Molecular Instruments | |
| **Chemicals, enzymes, and other reagents** | | |
| ProLong Gold Antifade Mountant | Invitrogen | P36930 |
| Vectashield | Vector Laboratories | H-1000-10 |
| Triton X-100 | Sigma-Aldrich | T9284 |
| Erioglaucine disodium salt | Sigma-Aldrich | 861146 |
| 16% Formaldehyde Solution (w/v) | ThermoFisher Scientific | 28908 |

| Reagent/resource | Reference or source | Identifier or catalog number |
| --- | --- | --- |
| **Software** | | |
| GraphPad Prism | https://www.graphpad.com/ | |
| Adobe Illustrator v. 27.2 | Adobe | |
| Leica LASX 3D Viewer | Leica | |
| Rstudio | Rstudio | |
| CellRanger v7.1.10 | 10x Genomics | |
| PANTHER 17.0 | Mi et al, 2013 | |
| Fiji v. 2.14.0 | ImageJ | |
| **Other** | | |
| Illumina Novaseq 6000 | Illumina | |
| S800Z Cell Sorter | Sony | |

## Fly husbandry and stocks

Embryos were collected on apple juice plates supplemented with wet yeast. Larvae were fed with standard sugar/yeast agar (SYA) food and kept at 25 °C. For amino acid starvation experiments, larvae were moved into 20% sucrose in PBS. In temperature shift experiments, newly hatched larvae were kept at 29 °C for 20 h, which is approximately equivalent to 24 h at 25 °C. For RNAi experiments, larvae were kept at 29 °C. Feeding assays were performed by adding ~0.1% erioglaucine disodium salt (Sigma) to SYA food. For EdU experiments, larvae were transferred to SYA food plates containing 0.2 mM EdU (ThermoFisher) 4 h prior to dissection.

The following lines were used: wor-GAL4 (Albertson et al, 2004), elav-GAL4 (Luo et al, 1994), grh-GAL4 (Chell and Brand, 2010), UAS-mCD8-GFP (Lee and Luo, 1999), tubGAL80$^{ts}$ (McGuire et al, 2003), tsh-GAL80 (gift from Gero Miesenboeck), UAS-myr-AKT (Stocker et al, 2002), UAS-PTEN (Huang et al, 1999), UAS-Kir2.1-GFP (Baines et al, 2001), UAS-TrpA1 (BDSC 26263), w$^{1118}$, UAS-higRNAi (BDSC 42000), UAS-mCD8-mCherry (BDSC 27391), UAS-mCherryRNAi (BDSC 35785), UAS-Dcr2 (Dietzl et al, 2007), UAS-haspRNAi #1 (VDRC, v30767), UAS-haspRNAi #2 (VDRC, v101250).

## Immunofluorescence

Larval brains were dissected in PBS and fixed in 4% formaldehyde in PBS for 20 min at RT. Dissected tissues were washed in 0.3% Triton in PBS (PBST) for 3 ×5 min at RT. The following primary antibodies were used: chicken anti-GFP (1:2000, Abcam ab13970), guinea pig anti-Dpn (Caygill and Brand, 2017) (1:5000), rabbit anti-pH3 (1:200, Millipore 06-570), rat anti-Wor (1:200, Abcam ab195173), mouse anti-Fas2 (1:100, DSHB 1D4), rabbit anti-CycA (1:10,000, gift from David Glover, rb270) (Whitfield et al, 1990), rabbit anti-RFP (1:10,000, Abcam, ab62341), rat anti-Elav (1:100, DSHB 7E8A10), guinea pig anti-Hig (Nakayama et al, 2014) and rat anti-Hasp (Nakayama et al, 2016) (1:1000, gift from Dr Chihiro Hama, Kyoto Sangyo University, Japan), rabbit anti-Rx (1:1000, gift from Uwe Walldorf) (Davis et al, 2003).

Antibodies were diluted in 0.3% PBST, and samples were incubated overnight at 4 °C. Samples were washed 3 ×15 min in 0/3% PBST at RT. Secondary antibodies (Alexa Fluor 405, 488, 546, 568, 633 (Life Technologies); DyLight 405 (Jackson Laboratories)) were diluted in PBST at 1:200 concentration and incubated overnight at 4 °C. Samples were mounted using Vectashield (Vector Laboratories).

## Hybridization chain reaction (HCR)

Following fixation in 4% formaldehyde in PBS for 20 min, larval brains were washed in PBS for 3 ×5 min at RT. PBS was replaced with Probe Hybridization Buffer (Molecular Instruments). Probes were added to the samples and incubated at 37 °C overnight. Samples were washed with Probe Wash Buffer (Molecular Instruments) for 4 ×15 min, followed by a 5-minute wash with 500 µl 50% probe wash buffer/50% 5×SSC-0.1% Triton and 2 ×5 min washed with 5×SSC-0.1% Triton. Hairpins were then added to the sample and incubated at RT in the dark overnight. Samples were washed with 5×SSC-0.1% Triton for 2 ×5 min, 2 ×30 min, and 1 ×5 min and then mounted in ProLong Gold Antifade Mountant (Invitrogen).

## Imaging and quantification

Images were acquired using a Leica SP8 upright confocal and a Nikon AX Inverted Confocal. Image processing and cell counting were performed using Fiji (v. 2.14.0). Figures were made using Adobe Illustrator (v. 27.2). 3D reconstruction was performed using the Leica LASX 3D Viewer software using maximum intensity projection images.

## Statistical analysis

Graphs were generated, and statistical analysis was performed using Prism (v. 9.5.1). Where applicable, a Mann–Whitney $U$ test was used, $P$ values and the number of biological replicates ($n$) are reported in the figure legends.

## scRNAseq sample preparation and data analysis

Stage 17 embryos (quiescence) and 24 h ALH (reactivated) larval brains were dissociated via mechanical and enzymatic dissociation. In order to enrich for neural cells, the neural-specific driver *wor*-GAL4 was used to express a membrane-bound GFP (UAS-mCD8-GFP) to facilitate fluorescence-activated cell sorting (FACS) on single-cell suspensions. Live (Propidum iodide-negative) and GFP+ cells were isolated using a S800Z Cell Sorter (Sony). Cells were then processed using the Chromium Next GEM Single Cell Library and Gel Bead Kit (v2 and v3.1) (10X Genomics). Sequencing was performed on a Novaseq 6000 (SP well) (Illumina) by the CRUK Genomics Facility.

FASTQ sequencing files were de-multiplexed and processed using CellRanger v7.1.10 (10X Genomics), and output files were analyzed using Seurat V4 (Hao et al, 2021). Cells with less than 200 and more than 6000 unique mRNA feature counts, and more than 5% mitochondrial counts, were excluded from analysis. The resulting datasets were analyzed with default parameters as described by the Satija lab (satijalab.org) using integrative analysis. A total of four replicates (2 quiescent, 2 reactivated) of scRNA-seq

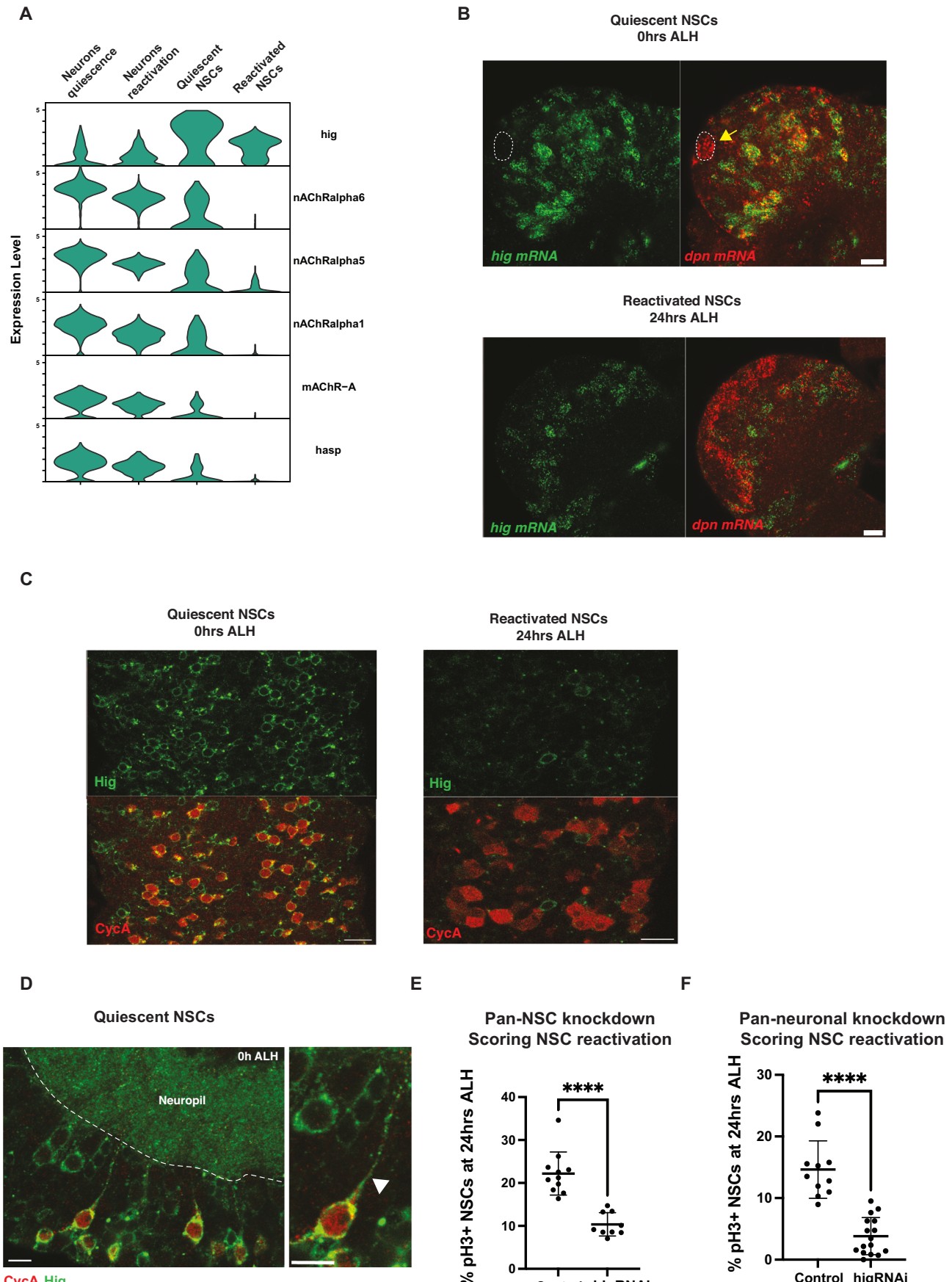

**Figure 5.  *hig* is required for timely NSC reactivation.**

(A) Cholinergic genes are upregulated during quiescence, including *hig* and its interactor *hasp* (B) *hig* mRNA is expressed in NSCs (marked by *dpn* mRNA in red) during quiescence (0 h ALH) and is downregulated upon reactivation (24 h ALH), yellow arrow and white dashed lines indicate a mushroom body NSC, which does not undergo quiescence and does not express *hig* (Fig. EV3), scale bars, 10 μm. (C) NSCs express Hig protein during quiescence (0 h ALH), but not after reactivation (24 h ALH), scale bars, 10 μm. (D) Hig is accumulates along the NSC projection during quiescence (0 h ALH), CycA in red marks $G_2$ NSCs, white arrow indicates Hig along an NSC projection, white dotted line indicates the neuropil boundary, scale bars, 5 μm, single confocal section. (E) *hig* knockdown in NSCs (*wor*-GAL4) impairs reactivation $n = 11$ (control, mCherryRNAi), $n = 9$ (higRNAi). (F) Pan-neuronal *hig* knockdown (*elav*-GAL4), $n = 11$ (control, mCherryRNAi), $n = 16$ (higRNAi). NSC reactivation is scored in the brain lobes. Mann–Whitney *U* test; ****$P < 0.0001$. Error bars indicate SD; the center is the mean. Source data are available online for this figure.

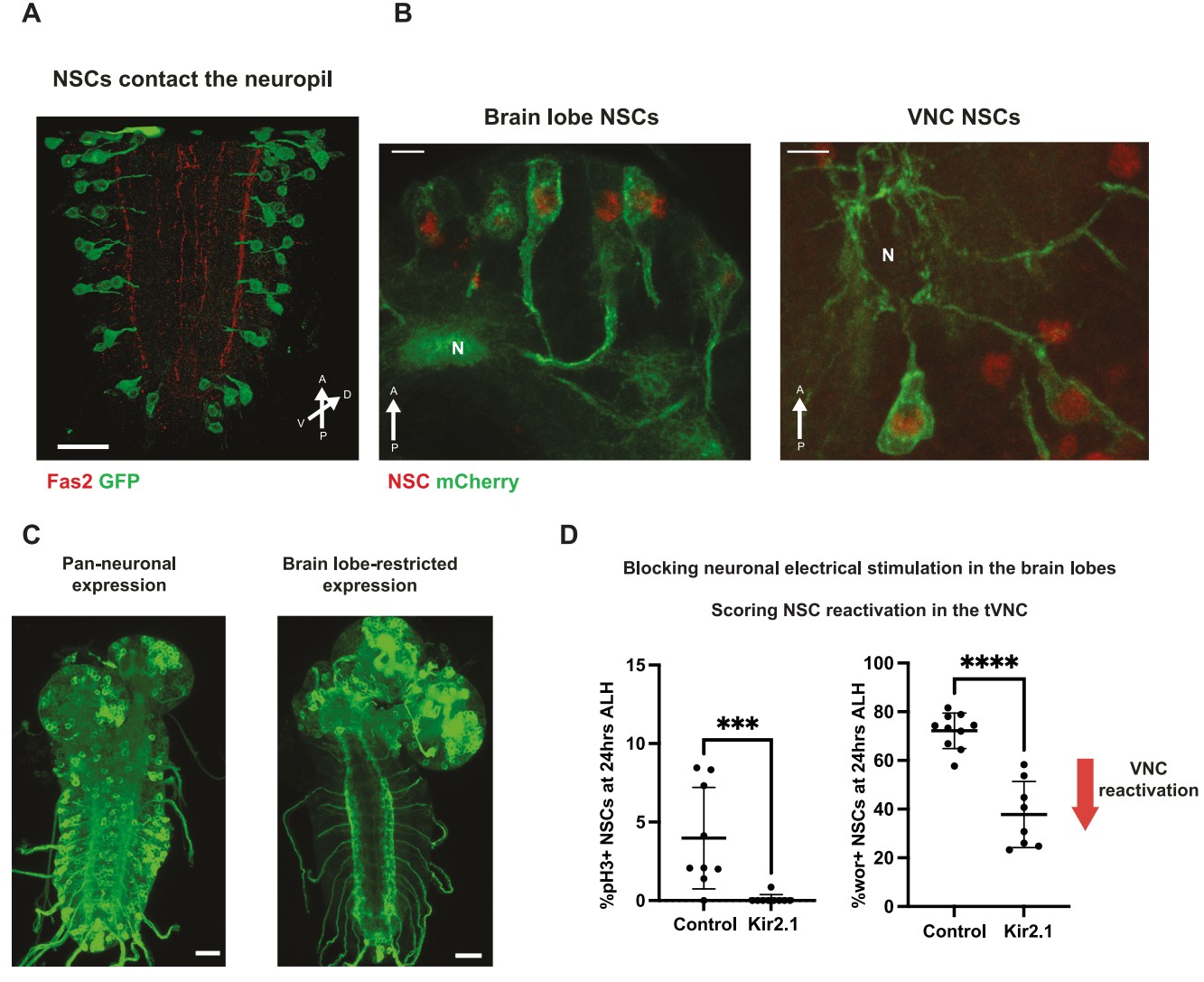

**A**

**NSCs contact the neuropil**

Fas2 GFP

**B**

**Brain lobe NSCs**

**VNC NSCs**

NSC mCherry

**C**

**Pan-neuronal expression**

**Brain lobe-restricted expression**

**D**

**Blocking neuronal electrical stimulation in the brain lobes**

**Scoring NSC reactivation in the tVNC**

***

****

VNC reactivation

**Figure 6.  Electrical inactivation of neurons impairs reactivation.**

(A) Quiescent NSCs extend a projection towards the neuronal tracts. Maximum intensity projection of the VNC; Fas2 (marking the descending neuronal tracts in the neuropil) in red, *wor*-GAL4 > mCD8-GFP in green. Scale bar, 20 μm. (B) Quiescent NSCs in the brain lobes and VNC contacting neuronal fibers. NSCs are marked by Dpn expression (red) and *grh*-GAL4 > mCD8-mCherry in green, Scale bars, 5 μm, N – neuropil, 18 h ALH. (C) Ectopic gene expression is restricted to the brain lobes using *elav*-GAL4, *tsh*-GAL80. Scale bar, 20 μm; mCD8-GFP in green at 24 h ALH. (D) Hyperpolarisation of brain lobe neurons and their descending tracts using *elav*-GAL4,*tsh*-GAL80 > UAS-Kir2.1-GFP, tubGAL80ts leads to impaired reactivation, and feeding is not impacted (Fig. EV5). Reactivation is scored in the tVNC at 24 h ALH; $n = 10$ (control, pH3), $n = 9$ (Kir2.1, pH3), *P* value = 0.0004; $n = 10$ (control, Wor), $n = 8$ (Kir2.1, Wor); Mann–Whitney *U* test; ***$P < 0.001$, ****$P < 0.0001$. Error bars indicate SD; and the center is the mean. Source data are available online for this figure.

**A**

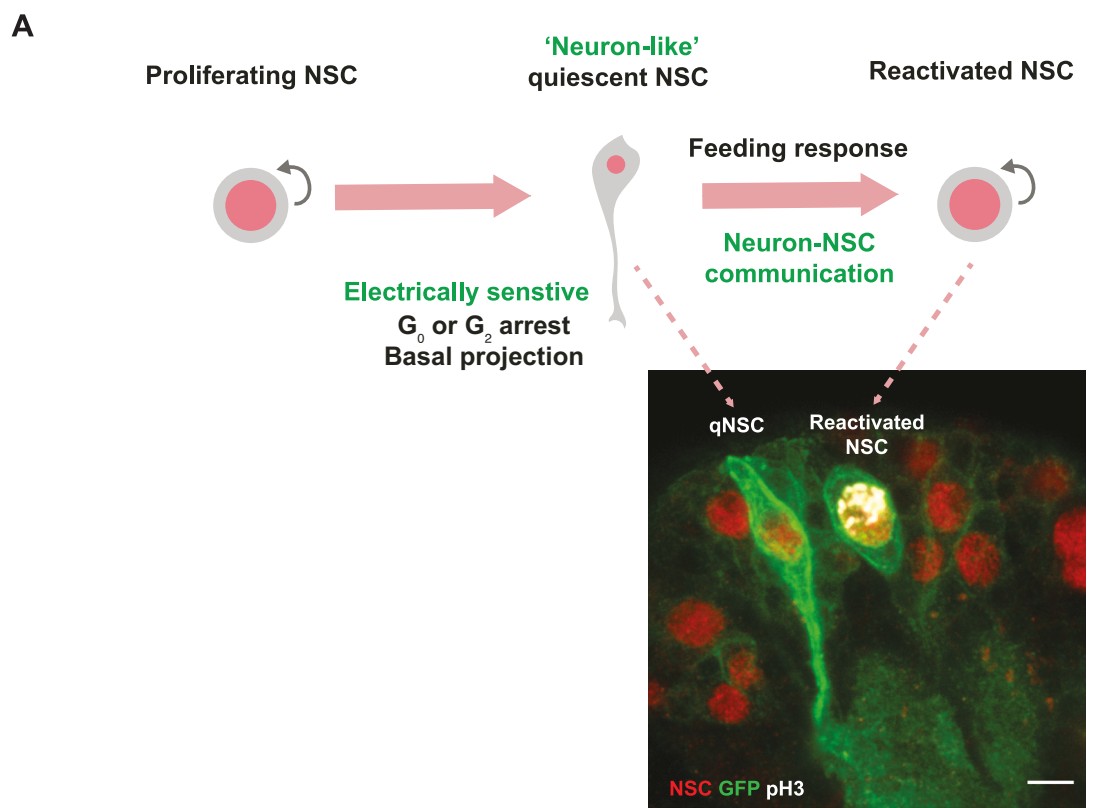

**B**

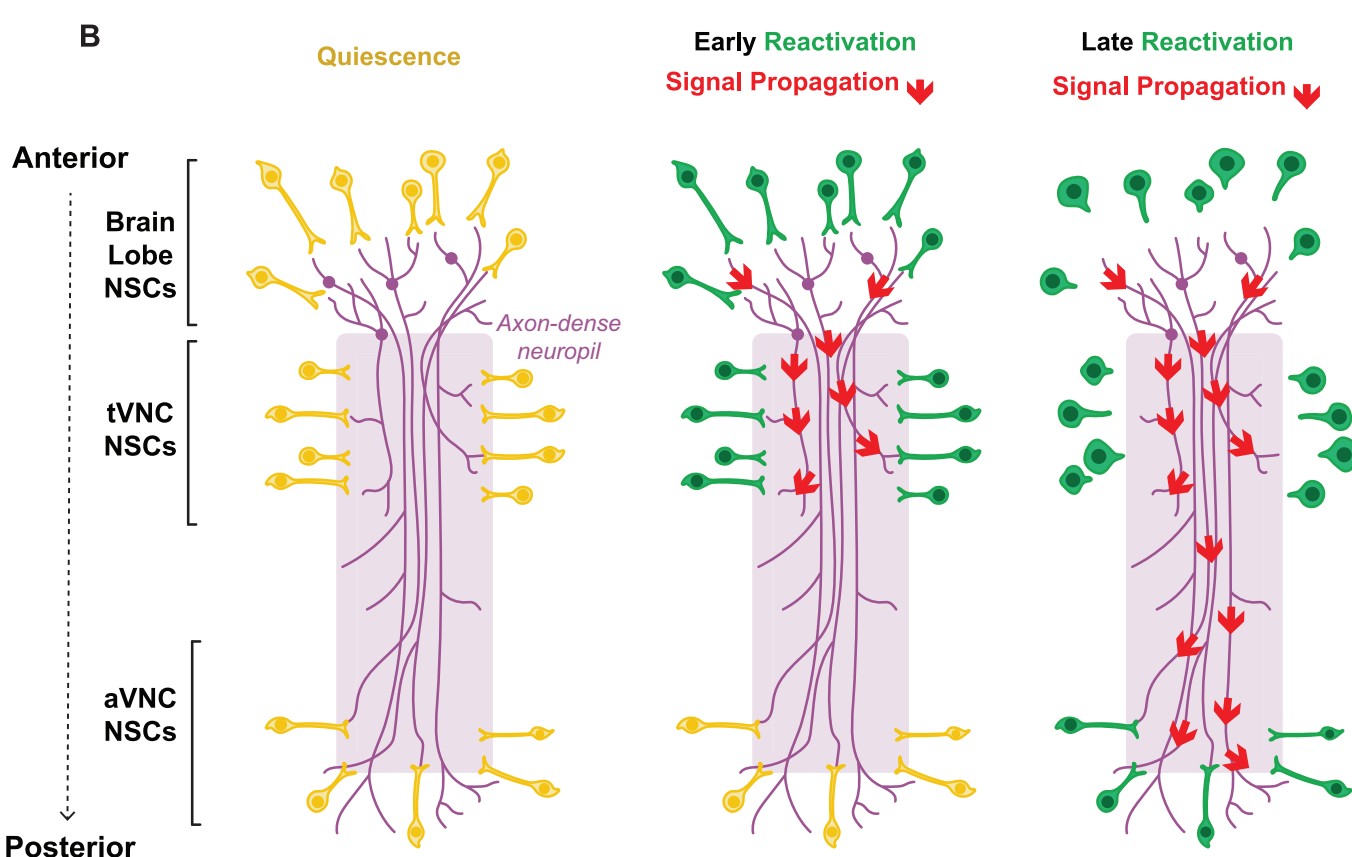

**Figure 7.   Quiescent NSCs adopt neuronal characteristics to coordinate reactivation via neuronal signaling.**

(**A**) Quiescent NSCs adopt a unique neuronal morphology which allows for communication with neurons within the neuropil. Proliferating NSCs pre- and post-quiescence do not maintain this morphology. The confocal image shows a qNSC and a pH3+ proliferating NSC at 16 h ALH. NSCs are marked by Dpn in red, and the membrane is marked using *wor*-GAL4 > mCD8-GFP. Scale bar, 5 µm. Our experiments suggest that the neuronal morphology allows for direct neuron-NSC communication. (**B**) NSCs reactivate in a spatially and temporally controlled manner, along the anterior–posterior axis. The descending neurons that project from the brain lobes to the ventral nerve cord enable the state of the NSCs in more anterior regions to be sensed by NSCs in the posterior region. Source data are available online for this figure.

produced a total of 3577 cells with a median number of (quiescent: 1250, 1528, reactivated: 4373) genes detected per cell.

We used the UCell package (Andreatta and Carmona, 2021) in R to generate a neuronal score based on the top 30 genes expressed in the mature neuronal clusters (i.e., nSyb+ pros-) during quiescence (stage 17). GO term enrichment analysis was performed using PANTHER 17.0 (Mi et al, 2013).

## Data availability

scRNAseq is available on GEO: GSE319955.

The source data of this paper are collected in the following database record: biostudies:S-SCDT-10_1038-S44318-026-00775-3.

## Peer review information

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

## Acknowledgements

We would like to thank the Gurdon Institute Imaging Facility (GIIF) for technical support. We thank Leia Judge and Anna Malkowska for help with the preparation of scRNA-seq samples. We would also like to thank Katarzynia Kania at the Cancer Research UK (CRUK) Cambridge Institute Genomics Core Facility for sample preparation and sequencing of samples. This work was funded by Wellcome Senior Investigator Award (103792), Wellcome Investigator Award (223111), and Royal Society Darwin Trust Research Professorship (RP150061) to AHB. JLYT was supported by a Wellcome Trust PhD Studentship (203798), and LO was funded by a Wellcome Trust PhD Studentship (097423). AHB acknowledges core funding to the Gurdon Institute from the Wellcome Trust (092096) and CRUK (C6946/A14492).

## Author contributions

**Laura-Yvonne Gherghina**: Conceptualization; Formal analysis; Validation; Investigation; Visualization; Methodology; Writing—original draft; Writing—review and editing. **Jocelyn LY Tang**: Conceptualization; Formal analysis; Validation; Investigation; Visualization; Methodology; Writing—original draft; Writing—review and editing. **Leo Otsuki**: Conceptualization; Formal analysis; Validation; Investigation; Visualization; Methodology; Writing—review and editing. **Leia Judge**: Visualization. **Andrea H Brand**: Conceptualization; Resources; Formal analysis; Supervision; Funding acquisition; Investigation; Methodology; Writing—original draft; Project administration; Writing—review and editing.

Source data underlying figure panels in this paper may have individual authorship assigned. Where available, figure panel/source data authorship is listed in the following database record: biostudies:S-SCDT-10_1038-S44318-026-00775-3.

## Disclosure and competing interests statement

The authors declare no competing interests.

# Expanded View Figures

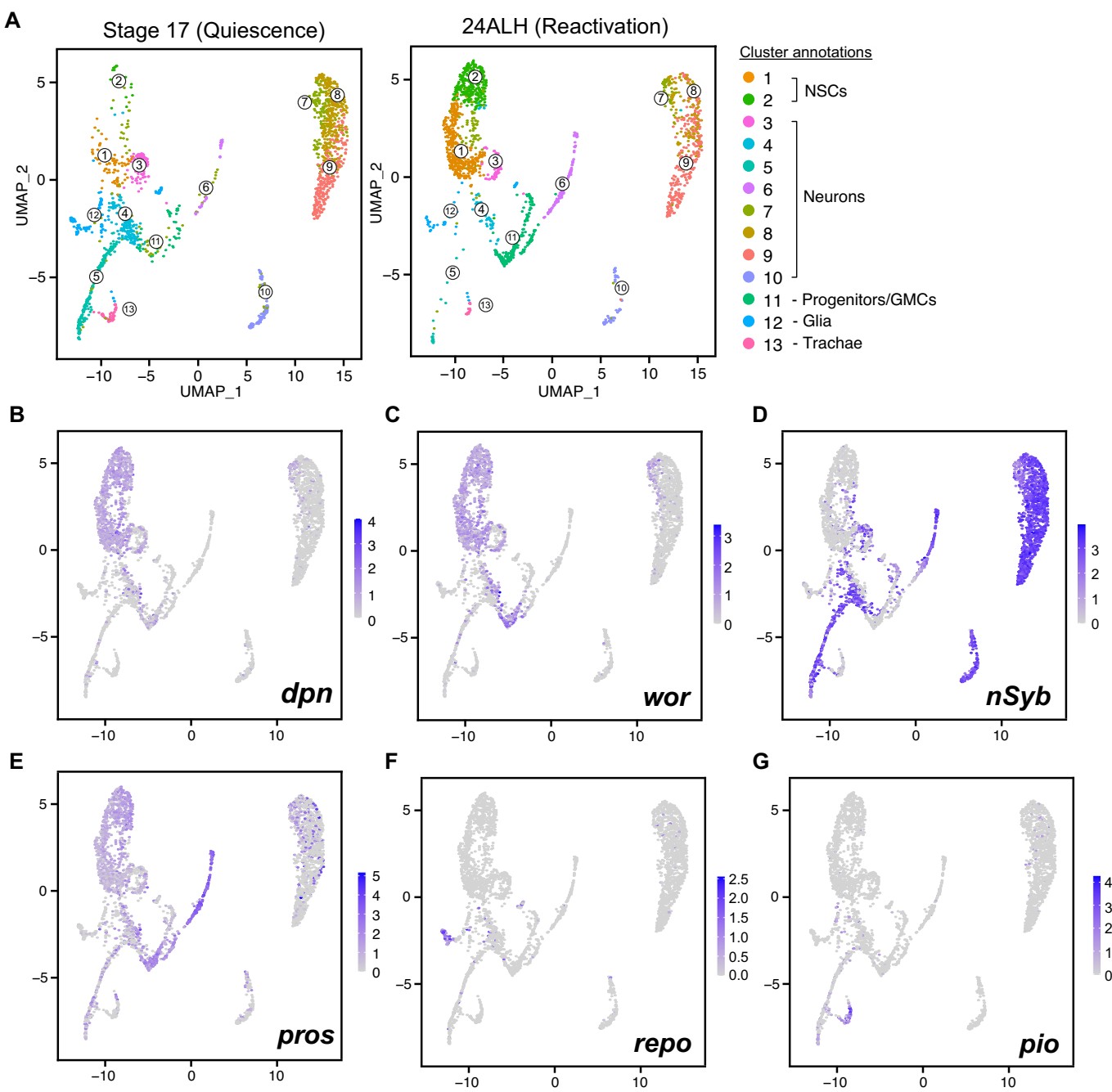

**Figure EV1.  Integrated scRNAseq datasets (quiescence and reactivation).**

(**A**) UMAPs show different clusters assigned and distribution in each timepoint. Cells of the central nervous system (CNS) were successfully isolated and sequenced, with the majority of cells being neurons (clusters 3–10 with neuronal *Synaptobrevin* (*nSyb*) expression), as well as smaller populations of glia (cluster 12, with *reversed polarity* (*repo*) expression), NSCs (clusters 1-2, with *deadpan* (*dpn*) and *wor* expression), a population of progenitors (progenitors (cluster 11, with *prospero* expression and absence of *dpn*) and trachae (cluster 13, *grh* and *piopio* expression). (**B**) *dpn* expression, specific to NSCs, (**C**) *wor* expression, NSC-specific, (**D**) nSyb expression, specific to neurons (*nSyb* positive, *dpn* negative cells), (**E**) *pros* expression, (**F**) *repo* expression, specific to glia (**G**), *pio* expression, specific to trachea.

**A**

**Expression of the top 30 genes in quiescent NSCs**

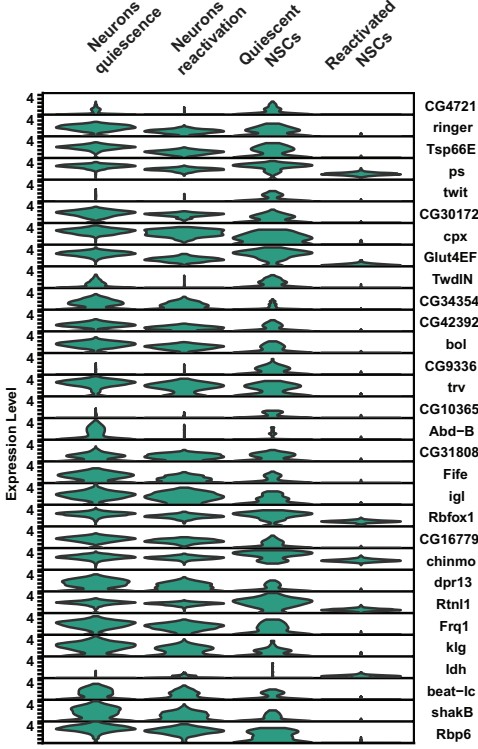

**B**

**Expression of the top 30 genes in neurons during quiescence**

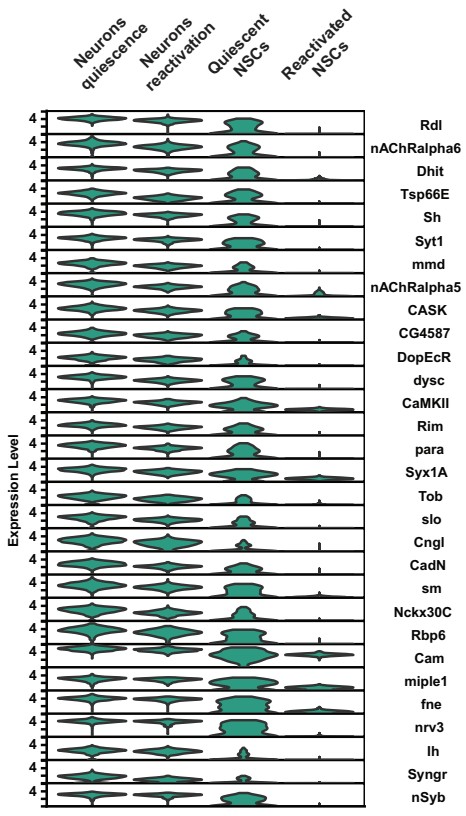

**Figure EV2. Expression of genes in NSCs from single-cell transcriptomic data.**

(A) Violin plots showing the average expression of the top 30 genes enriched in qNSCs. Expression levels of these 30 genes are shown in neurons (while NSCs are quiescent), qNSCs, neurons (while NSCs are reactivating), and reactivating NSCs. (B) Violin plots showing the average expression of the top 30 genes enriched in neurons. Expression levels of these 30 genes are shown in neurons (while NSCs are quiescent), qNSCs, neurons (while NSCs are reactivating), and reactivating NSCs.

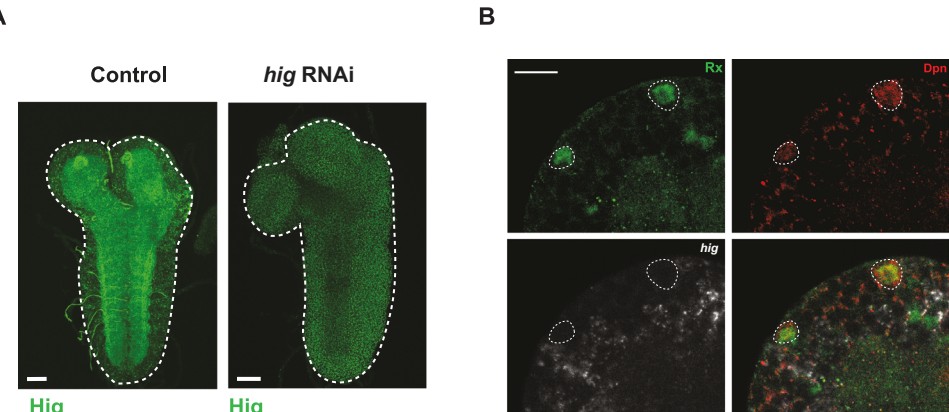

**A** Control | *hig* RNAi

Hig | Hig

**B** Rx | Dpn | *hig*

**Figure EV3.  Neuronal *hig* knockdown efficiency and lack of *hig* expression in mushroom body NSCs.**

(**A**) Efficiency of *elav*-GAL4>higRNAi, Control: *elav*-GAL4 > w[1118]. 0 h ALH at 29 °C, Scale bars – 20um. White dashed lines indicate the CNS outline. (**B**) *hig* is not expressed in mushroom body (MB) NSCs (Rx + ) at 0 h ALH, scale bars, 10 μm, white dashed lines indicate the MB NSC. Source data are available online for this figure.

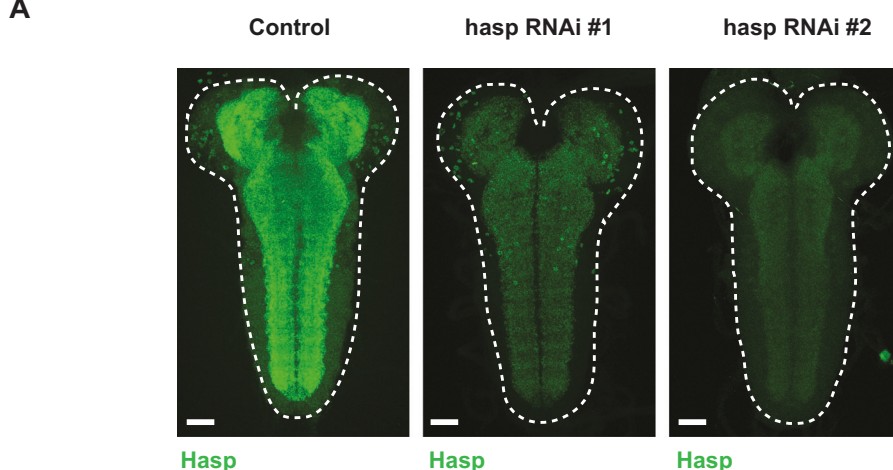

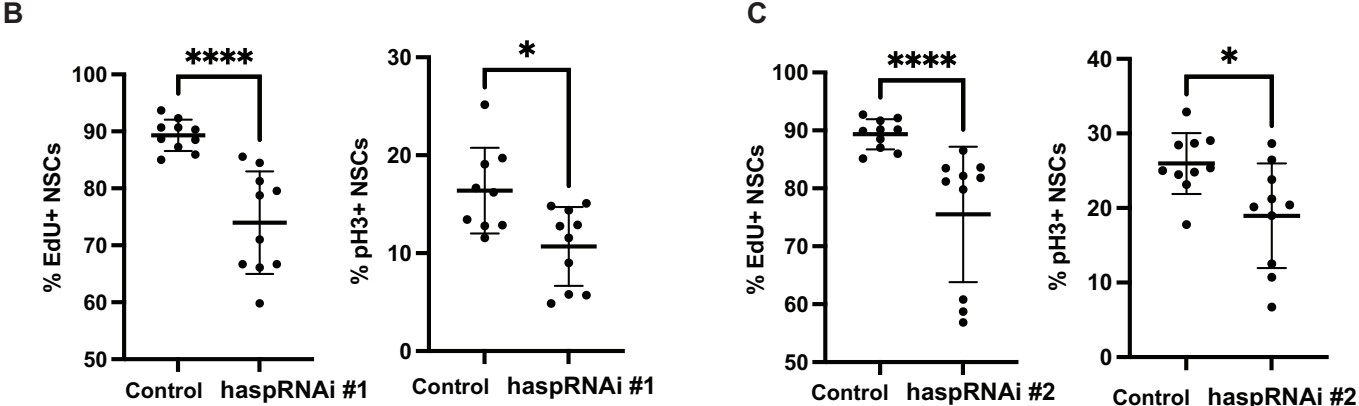

**Figure EV4.  Neuronal *hasp* knockdown impairs NSC reactivation.**

(A) Efficiency of two *hasp* RNAi constructs expressed using *elav*-GAL4 with UAS-Dcr2 in the genetic background, control is *elav*-GAL4 > w$^{1118}$, 24 h ALH at 29 °C. Scale bars, 20 μm. White dashed lines indicate the CNS outline. (B, C) *hasp* knockdown in neurons using *elav*-GAL4 leads to impaired NSC reactivation, quantified in the tVNC at 24 h ALH, *n* = 9 control (#1,pH3), *n* = 10 control (#1 EdU, #2)/haspRNAi#1/haspRNAi#2. *P* value (haspRNAi #1 pH3) = 0.0182, *P* value (haspRNAi #2 pH3) = 0.0185. ****P < 0.0001, Mann–Whitney *U* test; error bars indicate SD, and the center is the mean. Source data are available online for this figure.

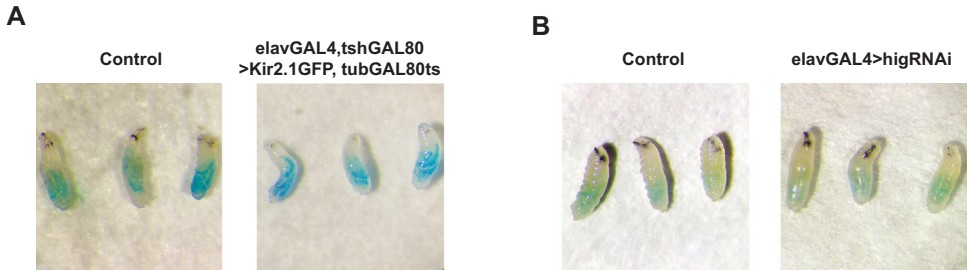

**A**  Control    elavGAL4,tshGAL80
>Kir2.1GFP, tubGAL80ts

**B**  Control    elavGAL4>higRNAi

**Figure EV5.  Nutritional intake upon neuronal hyperpolarisation.**

(**A**) Larvae continue to feed upon hyperpolarisation of brain lobe neurons after 0 h ALH (Control: *elav*-GAL4, *tsh*-GAL80 > UAS-mCD8-GFP, tubGAL80ts) and (**B**) RNAi-mediated knockdown of *hig* (Control: *elav*-GAL4 > UAS-mCherryRNAi). Images taken at 24 h ALH. Source data are available online for this figure.

