## [Peer Review File · The EMBO Journal]

Quiescent neural stem cells transiently become neuron-like to coordinate long-range reactivation

Laura-Yvonne Gherghina, Jocelyn L.Y. Tang, Leo Otsuki, Leia Judge, and Andrea H. Brand

Corresponding author: Andrea Brand (andrea.brand@nyulangone.org)

Review Timeline:

Submission Date:	9th Jul 25
Editorial Decision:	15th Aug 25
Revision Received:	17th Nov 25
Editorial Decision:	9th Feb 26
Revision Received:	2nd Mar 26
Accepted:	26th Mar 26

Editor: Daniel Klimmeck

Transaction Report:

Dear Dr Brand,

Thank you again for the submission of your manuscript (EMBOJ-2025-121853) to The EMBO Journal. Your study was assessed by two reviewers with expertise in adult neurogenesis and stem cell niche biology, whose comments are enclosed below.

As you will see from their comments, the referees acknowledge the potential interest and value of your results on reactivation of quiescent NSCs in the fly CNS in a temporally and spatially defined sequence and reliance of this pattern on non-cell-autonomous signaling between NSC populations and acquisition of neuron-like functional features. However, both referees express major reservations with your analysis pointing to potentially confounding aspects which need more rigorous evaluation to solidify the claims made (ref#1, pt.1). The first expert also points out that a functional role of the reactivated NSCs in long-range interactions is not sufficiently supported by the current data (ref#1, pt.5). Reviewer #2 states that technical robustness concerns (ref#2, pts.1-4) remain, which undermine the enthusiasm of this expert for the work. The referee also points out that the molecular basis for the proposed relay mechanism remains only prematurely addressed (ref#2, pts.6,7).

Further, the reviewers raise a number of issues related to the presentation of the findings, additional controls and improved methods annotation required, statistics applied and overall discussion of related literature, that would need to be conclusively addressed to achieve the level of robustness and clarity needed for The EMBO Journal.

Given the overall interest stated and broader angle of your results, we are able to invite you to revise your manuscript experimentally to address the referees' comments, along the lines sketched in your outline. I need to stress though that we do require strong support from the referees on a revised version of the study in order to move on to publication of the work. As to the open outcome of the revisional work and its evaluation I suggest keeping EMBO Reports in mind for this study as an alternative venue.

I would appreciate if you could contact me during the next weeks for exchange e.g. a video call to discuss your perspective on the comments and potential plan for revisions.

When submitting your revised manuscript, please carefully review the instructions below.

Please feel free to approach me any time should you have additional questions related to this.

Thank you for the opportunity to consider your work for publication.

I look forward to your revision.

Kind regards,

Daniel Klimmeck

Daniel Klimmeck, PhD
Senior Editor
The EMBO Journal

Instruction for the preparation of your revised manuscript:

- 1) a .docx formatted version of the manuscript text (including legends for main figures, EV figures and tables). Please make sure that the changes are highlighted to be clearly visible.
- 2) individual production quality figure files as .eps, .tif, .jpg (one file per figure).
- 3) a .docx formatted letter INCLUDING the reviewers' reports and your detailed point-by-point response to their comments. As

part of the EMBO Press transparent editorial process, the point-by-point response is part of the Review Process File (RPF), which will be published alongside your paper.

4) a complete author checklist, which you can download from our author guidelines ([https://wol-prod-cdn.literatumonline.com/pb-assets/embo-site/Author Checklist%20-%20EMBO%20J-1561436015657.xlsx](https://wol-prod-cdn.literatumonline.com/pb-assets/embo-site/Author%20Checklist%20-%20EMBO%20J-1561436015657.xlsx)). Please insert information in the checklist that is also reflected in the manuscript. The completed author checklist will also be part of the RPF.

6) It is mandatory to include a 'Data Availability' section after the Materials and Methods. Before submitting your revision, primary datasets produced in this study need to be deposited in an appropriate public database, and the accession numbers and database listed under 'Data Availability'. Please remember to provide a reviewer password if the datasets are not yet public (see <https://www.embopress.org/page/journal/14602075/authorguide#datadeposition>).

7) Our journal encourages inclusion of *data citations in the reference list* to directly cite datasets that were re-used and obtained from public databases. Data citations in the article text are distinct from normal bibliographical citations and should directly link to the database records from which the data can be accessed. In the main text, data citations are formatted as follows: "Data ref: Smith et al, 2001" or "Data ref: NCBI Sequence Read Archive PRJNA342805, 2017". In the Reference list, data citations must be labeled with "[DATASET]". A data reference must provide the database name, accession number/identifiers and a resolvable link to the landing page from which the data can be accessed at the end of the reference. Further instructions are available at .

8) At EMBO Press we ask authors to provide source data for the main and EV figures. Our source data coordinator will contact you to discuss which figure panels we would need source data for and will also provide you with helpful tips on how to upload and organize the files.

Numerical data can be provided as individual .xls or .csv files (including a tab describing the data). For 'blots' or microscopy, uncropped images should be submitted (using a zip archive or a single pdf per main figure if multiple images need to be supplied for one panel). Additional information on source data and instruction on how to label the files are available at .

9) We replaced Supplementary Information with Expanded View (EV) Figures and Tables that are collapsible/expandable online (see examples in <https://www.embopress.org/doi/10.15252/embo.201695874>). A maximum of 5 EV Figures can be typeset. EV Figures should be cited as 'Figure EV1, Figure EV2' etc. in the text and their respective legends should be included in the main text after the legends of regular figures.

11) For data quantification: please specify the name of the statistical test used to generate error bars and P values, the number (n) of independent experiments (specify technical or biological replicates) underlying each data point and the test used to calculate p-values in each figure legend. The figure legends should contain a basic description of n, P and the test applied. Graphs must include a description of the bars and the error bars (s.d., s.e.m.).

The revision must be submitted online within 90 days; please click on the link below to submit the revision online before 13th Nov 2025.

Referee #1:

Summary and opinion: This study investigates the mechanisms governing neural stem cell (NSC) reactivation following a period of quiescence in the developing *Drosophila* brain. The authors report that NSCs resume proliferation sequentially along the anteroposterior axis and seek to elucidate the molecular and cellular basis of this spatial pattern.

Using the powerful *Drosophila* genetic toolkit in combination with single-cell RNA sequencing, they propose that anterior NSCs (located in the brain) promote the reactivation of more posterior NSCs (located in the ventral nerve cord) through a neuronal relay mechanism. Particularly intriguing is their observation that quiescent NSCs appear to transiently adopt a neuron-like state, enabling communication via electrical signaling.

This work addresses an important and understudied question: how regional reactivation of NSCs is coordinated, which may be critical for maintaining tissue homeostasis in the aging or injured brain. The *Drosophila* model is employed effectively through elegant genetic manipulations. However, I have some concerns about whether the delayed reactivation of ventral nerve cord (VNC) NSCs might arise indirectly through altered nutritional signaling. This possibility should be carefully addressed to support the main conclusions of the study.

Overall, this is a novel and exciting study addressing a fundamental aspect of NSC biology. Addressing the points below should strengthen the manuscript and clarify the mechanistic basis of the proposed model.

Major Points

1. Wor-GAL4 Activity in Quiescent NSCs

The authors use wor-GAL4 to manipulate quiescent NSCs, yet wor-GAL4 is typically considered a marker of proliferating NSCs. Given that wor-GAL4 is used as a tool to evaluate quiescence exit, its activity during quiescence should be clarified. Is wor-GAL4 transcriptionally active but post-transcriptionally silenced in qNSCs? Please explain the rationale for using wor-GAL4 to target quiescent NSCs and provide supporting evidence if available.

2. Potential Confounding Effects of Nutritional Signaling

The authors convincingly manipulate brain NSCs or neurons to study their influence on posterior NSC reactivation. However, the manuscript does not address whether brain NSC reactivation is itself affected under these experimental conditions. If manipulations alter feeding behavior or interfere with insulin-producing cells, they could affect systemic nutritional signals, potentially explaining the delayed VNC NSC reactivation. In this situation, reactivation of brain NSCs should also be affected. Therefore, measuring brain NSC reactivation in all relevant conditions (particularly in Figures 5 and 6, where brain neurons are manipulated using elav-GAL4) would help validate the proposed relay mechanism and rule out indirect effects.

3. Clarification of Cell-Type Assignment in Single-Cell Analysis

In Figure 4d-e, the expression profiles are shown for different NSC states. Please clarify how these states were defined. Which clusters were assigned to quiescent vs. reactivated NSCs and neurons? Are quiescent and reactivated NSCs forming distinct clusters? What are the differentially expressed genes between clusters 1 and 2?

4. Figure 5B and 5C - Quantification and Annotation

These panels require quantification. Adding DAPI staining would help delineate cell boundaries and support the identification of NSCs. The observation that mushroom body neuroblasts lack hig expression is interesting and consistent with proliferating NBs losing neuronal markers, but the arrow pointing to a putative mushroom body neuroblast should be justified. What is the evidence that this is indeed an MB neuroblast?

5. Evidence Supporting Functional Role of Neuronal Genes in qNSCs

The functional evidence linking neuronal gene expression in qNSCs to inter-NSC communication is limited. Only a single UAS-hig-RNAi line is used, and its efficiency is not assessed. Since the authors report using a commercial anti-Hig antibody, please show evidence of antibody specificity and demonstrate that the RNAi line effectively reduces Hig expression. Furthermore, is this phenotype supported by targeting other components of the cholinergic signaling pathway (e.g., hasp, nAChRs)? These validations are required to support the idea that NSCs and neurons may relay signals via a synaptic mechanism.

Minor Points

1. Please describe briefly in the main text the genetic strategy used to specifically manipulate qNSCs in the VNC.

2. Panel annotations are incomplete:

o Figure 2a: Please indicate the age (ALH).

o Figure 4a: Please include genotype, antibodies used, and ALH.

3. The reference to "(Fig 5b, S1)" should be corrected to "(Fig 4b, S1)".

4. *worniu* RNA is detected in both quiescent and reactivated NSCs in the single-cell RNA-seq dataset, which may confuse readers. Since *Wor* is associated with active NSCs, please clarify why it is also expressed in qNSCs. This ties into major point #1 and warrants a clear explanation.

5. Improve image contrast in Figures 5b, 5c, 6a, and 6b to enhance visibility of labeled cells and structures.

6. Please provide references for all commercial antibodies used.

Referee #2:

This study investigates the temporal-spatial reactivation of quiescent neural stem cells (NSCs) in *Drosophila* during early larval development, with a particular focus on stem-cell-to-neuron communication mechanisms. Prior work has shown that NSC reactivation is primarily driven by insulin signaling from larval feeding or by neuronal input. Gherghina et al. propose that this process progresses along the anterior-posterior axis via a novel form of communication from anterior, neuron-like NSCs to projection neurons, which in turn reactivates posterior NSCs.

Their data support the idea that quiescent NSCs express neurotransmitter receptors-features lost upon reactivation-paralleling observations in adult mammalian NSCs (e.g., Shin et al., *Cell Stem Cell*, 2015). The study introduces the concept that anterior NSCs may depolarize and signal through projection neurons to coordinate posterior NSC reactivation. This concept is intriguing and, if substantiated, could expand our understanding of NSC circuitry and activation across species.

However, key conceptual and technical issues remain unresolved. Although the study raises compelling hypotheses, there are more unanswered questions than mechanistic insights at this stage. Additional experiments and clarifications are necessary before publication.

Major Concerns

1. The quantification of reactivated posterior NSCs using %pH3+ at 24h post-hatching shows significant variability across experiments. Control values range from 3% to 23% depending on the figure, making comparisons difficult. A clearer explanation of this variability and normalization strategies is essential.

2. Some figure panels lack clarity in labeling. For example, in Figure 5, "NSC" is labeled in red but does not correspond to an antibody. Higher magnification images and quantification of colocalization would improve interpretability.

3. Figure 5c images are very dim and difficult to interpret. In addition, quantification is lacking in several key panels. For example, Figures 5e and 5f show unequal sample sizes ($n = 5$ vs. 16), affecting statistical confidence.

4. The schematic in Figure 7 implies morphological changes during quiescence exit, but these were not presented or quantified in the manuscript.

5. The manuscript emphasizes cholinergic receptor expression in NSCs (implying input from neurons), but it remains speculative how anterior NSCs communicate with projection neurons. Are they releasing neurotransmitters? More discussion of potential ligand-receptor interactions or candidate pathways (from scRNA-seq data) is needed.

6. The mechanism by which signals are conveyed from anterior NSCs to posterior NSCs remains vague. Given the prolonged time course (~24h), neurotransmission alone seems insufficient. Are these axons transmitting trophic factors or transporting signaling molecules?

7. The downstream receptors or mechanisms in posterior NSCs are not addressed. Genetic or pharmacological screens could help elucidate these targets.

Minor concerns:

1. The distinction between G0 and G2 quiescent NSCs is introduced early, but not further explored. If these subtypes differ in reactivation potential or mechanisms, this should be clarified or omitted.

2. The study concludes that both anterior NSC activation and neuronal activity are required for posterior NSC reactivation. A potential mechanism-such as axonal uptake and release of factors-should be explored further.

3. The title, abstract, and early text do not make it sufficiently clear that this study is performed in *Drosophila*. This is crucial for reader context and should be explicitly stated.

4. The manuscript title suggests that NSCs function as neurons. However, these NSCs do not exhibit hallmarks of neuronal identity such as action potentials, synaptic vesicles, or post-synaptic structures. This claim should be toned down or reworded to reflect a more accurate neuronal-like or receptor-expressing phenotype.

EMBOJ-2025-121853; referee reports

Referee #1:

Summary and opinion: This study investigates the mechanisms governing neural stem cell (NSC) reactivation following a period of quiescence in the developing *Drosophila* brain. The authors report that NSCs resume proliferation sequentially along the anteroposterior axis and seek to elucidate the molecular and cellular basis of this spatial pattern.

Using the powerful *Drosophila* genetic toolkit in combination with single-cell RNA sequencing, they propose that anterior NSCs (located in the brain) promote the reactivation of more posterior NSCs (located in the ventral nerve cord) through a neuronal relay mechanism. Particularly intriguing is their observation that quiescent NSCs appear to transiently adopt a neuron-like state, enabling communication via electrical signaling.

This work addresses an important and understudied question: how regional reactivation of NSCs is coordinated, which may be critical for maintaining tissue homeostasis in the aging or injured brain. The *Drosophila* model is employed effectively through elegant genetic manipulations. However, I have some concerns about whether the delayed reactivation of ventral nerve cord (VNC) NSCs might arise indirectly through altered nutritional signaling. This possibility should be carefully addressed to support the main conclusions of the study.

Overall, this is a novel and exciting study addressing a fundamental aspect of NSC biology. Addressing the points below should strengthen the manuscript and clarify the mechanistic basis of the proposed model.

Major Points

1. Wor-GAL4 Activity in Quiescent NSCs

The authors use wor-GAL4 to manipulate quiescent NSCs, yet *worniu* is typically considered a marker of proliferating NSCs. Given that wor-GAL4 is used as a tool to evaluate quiescence exit, its activity during quiescence should be clarified. Is *worniu* transcriptionally active but post-transcriptionally silenced in qNSCs? Please explain the rationale for using wor-GAL4 to target quiescent NSCs and provide supporting evidence if available.

Our apologies for this omission, we are happy to clarify. The gene, *worniu*, is active in both proliferating and quiescent NSCs. However, the Worniu protein is only expressed in proliferating NSCs (described in Lai and Doe, 2014, *eLife* & Otsuki and Brand 2018, *Science*). As a result, we can use wor-GAL4 to drive expression in quiescent NSCs, and the appearance of Worniu protein to identify proliferating NSCs. We have previously used wor-GAL4 to drive Targeted DamID in proliferating and quiescent NSCs, which revealed crucial regulators of NSC quiescence, such as Tribbles (Otsuki and Brand 2018, *Science*). We are presently studying the post-transcriptional regulation of Worniu.

We have clarified this in the text, by adding: *'We assessed reactivation based on expression of the protein Worniu (Wor). Whilst the wor transcript is expressed in NSCs*

throughout development, Wor protein is only present in NSCs upon reactivation, prior to mitosis (Otsuki & Brand, 2018).'

2. Potential Confounding Effects of Nutritional Signaling

The authors convincingly manipulate brain NSCs or neurons to study their influence on posterior NSC reactivation. However, the manuscript does not address whether brain NSC reactivation is itself affected under these experimental conditions. If manipulations alter feeding behavior or interfere with insulin-producing cells, they could affect systemic nutritional signals, potentially explaining the delayed VNC NSC reactivation. In this situation, reactivation of brain NSCs should also be affected. Therefore, measuring brain NSC reactivation in all relevant conditions (particularly in Figures 5 and 6, where brain neurons are manipulated using elav-GAL4) would help validate the proposed relay mechanism and rule out indirect effects.

To ensure that feeding behaviour was not affected, we performed feeding assays with blue food and found that larvae are still able to feed after hyperpolarisation of brain lobe neurons or knockdown of *hig* using elav-GAL4. We have included these experiments in a supplementary figure and below.

Regarding the secretion of insulin by the insulin-producing cells, this does not induce neural stem cell reactivation. Insulin-producing cells secrete insulin-like peptides (ILPs) into the haemolymph, but larval NSCs are not responsive to systemic ILPs (Chell and Brand 2010, Cell).

3. Clarification of Cell-Type Assignment in Single-Cell Analysis

In Figure 4d-e, the expression profiles are shown for different NSC states. Please clarify how these states were defined. Which clusters were assigned to quiescent vs. reactivated NSCs and neurons? Are quiescent and reactivated NSCs forming distinct clusters? What are the differentially expressed genes between clusters 1 and 2?

The scRNA-seq datasets for quiescent and reactivated NSCs were performed separately from stage 17 embryos (quiescence) and 24h ALH (reactivated) brains (please see methods for more details) and are shown together in Figure 4d (left and right, respectively). The clusters were generated by unsupervised clustering, using parameters suggested by the Seurat package (using elbow plots to determine the number of principle components appropriate for generating the plots in the figure). We performed integrative analysis on the two datasets (quiescence and

reactivation), which were shown separately in Supplementary figure 1. This has now been clarified in the methods.

The analysis was performed using all NSCs (i.e. both clusters) and we now have included a list of differentially expressed genes between NSCs clusters 1 and 2, but we did not find relevant differences between these clusters. We also clarified which neuronal clusters were used for the analysis; we excluded any nSyb+ clusters that also expressed *prospero* (*pros*) as this combination of markers indicates immature neurons.

4. Figure 5B and 5C - Quantification and Annotation

These panels require quantification. Adding DAPI staining would help delineate cell boundaries and support the identification of NSCs. The observation that mushroom body neuroblasts lack *hig* expression is interesting and consistent with proliferating NBs losing neuronal markers, but the arrow pointing to a putative mushroom body neuroblast should be justified. What is the evidence that this is indeed an MB neuroblast?

We used *dpn* mRNA to mark NSCs (by HCR), and distinguished mushroom body NSCs

by their location and cell size. During quiescence, we found that *hig* mRNA is present in all *dpn+* cells. We have added an Rx stain (Figure S3), which specifically marks MB NSCs, to show that *hig* mRNA is absent in MB NSCs, which never undergo quiescence. We have improved the contrast of the images in Figure 5 and included Hig immunostaining during quiescence and after NSC reactivation to further show its downregulation in proliferating NSCs.

b

5. Evidence Supporting Functional Role of Neuronal Genes in qNSCs

The functional evidence linking neuronal gene expression in qNSCs to inter-NSC communication is limited. Only a single UAS-*hig*-RNAi line is used, and its efficiency is not assessed. Since the authors report using a commercial anti-Hig antibody, please show evidence of antibody specificity and demonstrate that the RNAi line effectively reduces Hig expression. Furthermore, is this phenotype supported by

targeting other components of the cholinergic signaling pathway (e.g., hasp, nAChRs)? These validations are required to support the idea that NSCs and neurons may relay signals via a synaptic mechanism.

We observed reduced Hig levels resulting from *hig* RNAi expression, we have added this in Figure S3 and below.

We also knocked down hasp using *elav-GAL4* and found that NSC reactivation was impaired, providing further functional evidence for this mechanism. We have included these data in the supplementary materials (Figure S4).

The Hig and Hasp antibodies (used in Nakayama et al. 2016, JNeurosci) were a kind gift from C.Hama, Kyoto Sangyo University, we have clarified this in the methods.

Minor Points

1. Please describe briefly in the main text the genetic strategy used to specifically manipulate qNSCs in the VNC.

We have added a brief description to the text: '*We generated a genetic tool to restrict GAL4-mediated gene expression to NSCs in the brain lobes (Fig. 2a) by combining the NSC-specific driver wor-GAL4 with tsh-GAL80, which prevents GAL4 expression in the VNC (Simpson, 2016).*'

2. Panel annotations are incomplete:

o Figure 2a: Please indicate the age (ALH).

These were taken at 24hrs ALH, we have clarified this.

o Figure 4a: Please include genotype, antibodies used, and ALH.

We have included this.

3. The reference to "(Fig 5b, S1)" should be corrected to "(Fig 4b, S1)".

We have corrected this.

4. wor¹ RNA is detected in both quiescent and reactivated NSCs in the single-cell RNA-seq dataset, which may confuse readers. Since Wor is associated with active NSCs, please clarify why it is also expressed in qNSCs. This ties into major point #1 and warrants a clear explanation.

We have clarified that wor mRNA is present in NSCs, but it is only translated in reactivated NSCs.

5. Improve image contrast in Figures 5b, 5c, 6a, and 6b to enhance visibility of labeled cells and structures.

We have improved the contrast, apologies.

6. Please provide references for all commercial antibodies used.

We have added references for all reagents used.

Referee #2:

This study investigates the temporal-spatial reactivation of quiescent neural stem cells (NSCs) in *Drosophila* during early larval development, with a particular focus on stem-cell-to-neuron communication mechanisms. Prior work has shown that NSC reactivation is primarily driven by insulin signaling from larval feeding or by neuronal input. Gherghina et al. propose that this process progresses along the anterior-posterior axis via a novel form of communication from anterior, neuron-like NSCs to projection neurons, which in turn reactivates posterior NSCs.

Their data support the idea that quiescent NSCs express neurotransmitter receptors/features lost upon reactivation—paralleling observations in adult mammalian NSCs (e.g., Shin et al., Cell Stem Cell, 2015). The study introduces the concept that anterior NSCs may depolarize and signal through projection neurons to coordinate posterior NSC reactivation. This concept is intriguing and, if substantiated, could expand our understanding of NSC circuitry and activation across species.

However, key conceptual and technical issues remain unresolved. Although the study raises compelling hypotheses, there are more unanswered questions than mechanistic insights at this stage. Additional experiments and clarifications are necessary before publication.

Major Concerns

1. The quantification of reactivated posterior NSCs using %pH3+ at 24h post-hatching shows significant variability across experiments. Control values range from 3% to 23% depending on the figure, making comparisons difficult. A clearer explanation of this variability and normalization strategies is essential.

pH3 marks proliferating cells only in M-phase and, as such, the readout is variable. To supplement these data, we also used Wor expression to quantify reactivation in Figures 1,3 and 6. We note that for the UAS-myr-Akt experiment in Figure 2c, the starved control shows no reactivation, such that pH3 staining is sufficient to demonstrate the phenotype. For the UAS-PTEN experiment in Figure 2b, we have added quantification of Wor expression.

2. Some figure panels lack clarity in labeling. For example, in Figure 5, "NSC" is labeled in red but does not correspond to an antibody. Higher magnification images and quantification of colocalization would improve interpretability.

In Figure 5, NSCs are labelled by in situ hybridisation to *dpn* mRNA, as specified in the

figure legend. We have improved the contrast and added immunolabelling to further show that Hig is present in all NSCs specifically during quiescence.

3. Figure 5c images are very dim and difficult to interpret. In addition, quantification is lacking in several key panels. For example, Figures 5e and 5f show unequal sample sizes ($n = 5$ vs. 16), affecting statistical confidence.

We have replaced Figure 5c with clearer panels and have performed additional replicates of the *wor-GAL4> higRNAi* experiment to improve statistical confidence and also for Fig 3b.

4. The schematic in Figure 7 implies morphological changes during quiescence exit, but these were not presented or quantified in the manuscript.

Figure 4a shows quiescent NSCs with projections (middle panel) and proliferating NSCs lacking projections together with their progeny (right hand panel). This is in line with previous publications in which NSC projections are seen only in qNSCs, as reported previously (for example, Chell and Brand 2010, Cell). We have added an image to Figure 7 showing a quiescent NSC on the left, with a projection but no pH3 staining, and a reactivated NSC on the right, without a projection and staining for pH3 (white); NSCs stained with anti-Dpn in red, cell membranes marked in green.

5. The manuscript emphasizes cholinergic receptor expression in NSCs (implying input from neurons), but it remains speculative how anterior NSCs communicate with projection neurons. Are they releasing neurotransmitters? More discussion of potential ligand-receptor interactions or candidate pathways (from scRNA-seq data) is needed.

While we focused on cholinergic signalling, we cannot rule out other mechanisms, as the scRNA-seq data shows that qNSCs also express other receptors. To investigate further, we are exploring GAL4 drivers to mark and manipulate specific neuronal circuits, an exciting future direction.

We have included data showing that in addition to *hig*, the *hig* anchoring scaffold protein, *hasp*, is also required for NSC reactivation (Figure S4). This suggests that NSC-neuron communication may occur through synapse-like mechanisms, a point which we have now further developed in the Discussion.

6. The mechanism by which signals are conveyed from anterior NSCs to posterior NSCs remains vague. Given the prolonged time course (~24h), neurotransmission alone seems insufficient. Are these axons transmitting trophic factors or transporting signaling molecules?

NSCs reactivate progressively over 24hrs and, as shown in Figure 1, reactivation occurs at different timepoints depending upon the specific spatial niche. 24h is the time period required for the entire population of NSCs in the VNC to complete reactivation, as opposed to individual NSCs requiring 24h to reactivate.

7. The downstream receptors or mechanisms in posterior NSCs are not addressed. Genetic or pharmacological screens could help elucidate these targets.

We agree that a screen would be useful and a good future direction. However, we note that the non-autonomous control of NSC reactivation is a previously undescribed phenomenon which we believe opens up exciting future avenues of

investigation in the field.

Minor concerns:

1. The distinction between G0 and G2 quiescent NSCs is introduced early, but not further explored. If these subtypes differ in reactivation potential or mechanisms, this should be clarified or omitted.

We make the point that the timing of anterior to posterior reactivation cannot be accounted for solely by the difference in G2 and G0 reactivation times. While G2 NSCs

reactivate before G0 (Otsuki and Brand 2018, Science), this is insufficient to explain anterior NSCs reactivating earlier than posterior NSCs. Including these data is important to link our findings to prior knowledge in the field.

2. The study concludes that both anterior NSC activation and neuronal activity are required for posterior NSC reactivation. A potential mechanism-such as axonal uptake and release of factors-should be explored further.

We are investigating this further, but it is beyond the scope of this manuscript.

3. The title, abstract, and early text do not make it sufficiently clear that this study is performed in *Drosophila*. This is crucial for reader context and should be explicitly stated.

We have made it more explicit in the abstract and it is also specified in the introduction.

4. The manuscript title suggests that NSCs function as neurons. However, these NSCs do not exhibit hallmarks of neuronal identity such as action potentials, synaptic vesicles, or post-synaptic structures. This claim should be toned down or reworded to reflect a more accurate neuronal-like or receptor-expressing phenotype.

We have changed 'neuronal' to 'neuronal-like' in the title.

Dear Dr Brand,

Thank you for submitting your revised manuscript (EMBOJ-2025-121853R) to The EMBO Journal, as well for your patience with our feedback. Your amended study was sent back to the referees for their scientific reassessment, and we have received reports from both, which I enclose below. As you will see, the reviewers state that the work has been substantially enhanced by the revisions and they are now broadly in favour of publication, pending minor amendments.

Thus, we are pleased to inform you that your manuscript has been accepted in principle for publication in The EMBO Journal.

Please carefully consider the remaining minor points raised by referee #1 by adding complementary data or revising the manuscript text and discussion of the findings where appropriate.

Also, we now need you to take care of a number of issues related to formatting and data presentation as detailed below, which should be addressed at re-submission.

Please submit a revised version of the manuscript using the link enclosed below, addressing the advisor's comments.

As you might have seen on our web page, every paper at the EMBO Journal includes a 'Synopsis', displayed on the html and freely accessible to all readers. The synopsis includes a 'model' figure as well as 2-5 one-short-sentence bullet points that summarize the article. I would appreciate if you could provide this figure and the bullet points.

Thank you again for giving us the chance to consider your manuscript for The EMBO Journal, I look forward to hearing from you and receiving your final revised version of the manuscript.

Best regards,

Daniel Klimmeck

>> Please add up to five keywords to your study.

>> Author Contributions: Remove the author contributions information from the manuscript text. Note that CRediT has replaced the traditional author contributions section as of now because it offers a systematic machine-readable author contributions format that allows for more effective research assessment. and use the free text boxes beneath each contributing author's name to add specific details on the author's contribution.

More information is available in our guide to authors.
<https://www.embopress.org/page/journal/14602075/authorguide>

>> Adjust the title of the 'Competing Interests' section to 'Disclosure and Competing Interests Statement'.

>> Add a separate 'Statistical Analysis' section to the Methods part, detailing the algorithms and statistical tests applied.

>> Correct the order of the manuscript sections as follows: Abstract / Keywords / Introduction / Results / Discussion / Methods / Data Availability / Acknowledgements / Disclosure and Competing Interests Statement / References / Main Figure Legends / Tables / Expanded View Figure Legends.

>> Figures in separate files: Figures should be removed from the manuscript text and uploaded as individual, high resolution figure files. Legends should be placed after the References.

>> Figure callouts: Please ensure that the figures and panels are called out in sequential order. Currently, Fig 6A is called out before Fig. 5.

>> Funding: please merge with Acknowledgements, Wellcome Trust PhD Studentship (097423) is missing in our system, please also add Gurdon Institute from the Wellcome Trust (092096) and CRUK (C6946/A14492) to the funders list in our system if appropriate.

>> References: adjust the reference format to EMBO Journal format, alphabetical order and ten author names listed before et al. .

>> Data availability section: please ensure that the data are made publicly accessible. Add hyperlinks to the dataset.

>> Author checklist: Correct the Data Availability part accordingly.

>> Please provide source data for the study as to the earlier separate request e-mail by our office team. Source data should be uploaded as one (zipped) file per figure.

>> Add a Reagents and Tools table to the Methods section, as a separate file using the existing template in the Guide For Authors, listing key reagents, experimental models, software and relevant equipment.

>> Consider additional changes and comments from our production team as indicated below:

- DAS:

1. Please note that the accession ID/code for the "GEO database" is not provided in the data availability statement.
2. Please note that reviewer access code for "GEO" dataset is not provided in the data availability statement.

- Figure legends:

1. Please note that the titles for figure 5; supplementary figures 3; 4; 5 are missing in the manuscript. This needs to be rectified.
2. Please note that the exact p values are not provided in the legends of figures 2b, c; 3b, c; 5e, f; 6d; supplementary figures 4b, c
3. Please note that information related to n is missing in the legends of figures 4d; 5a; supplementary figures 2a, b; 4b, c
4. Please note that the measure of center for the error bars needs to be defined in the legends of figures 1b; 2b, c; 3b, c; 5e, f; 6d; supplementary figures 4b, c
5. Please note that the scale bar is missing for figures 6a
6. Please note that the white dashed area is not defined in the legend of figure 2a; 5b; supplementary figures 3a, b; 4a. This needs to be rectified.

Referee #1:

The authors have satisfactorily addressed most of my concerns. The manuscript is of high quality, and the findings are novel and highly interesting. I still have one comment that should be addressed prior to publication. The authors added a new Figure 5C demonstrating that Hig is specifically expressed in quiescent NSCs and displays a cortical localization. However, this cortical localization is not apparent in Figure 5D, and the authors' claim that Hig "accumulates along the NSC projection" in qNSCs is not fully convincing, as the Hig-positive puncta appear to be randomly distributed. Could the authors clarify why Hig subcellular localization appears to differ between Figures 5C and 5D?

Referee #2:

The authors have addressed all of my previous concerns through additional analyses, clarifications, and improved presentation of the data. I am satisfied with the responses and recommend acceptance of the manuscript.

Referee #1:

The authors have satisfactorily addressed most of my concerns. The manuscript is of high quality, and the findings are novel and highly interesting. I still have one comment that should be addressed prior to publication. The authors added a new Figure 5C demonstrating that Hig is specifically expressed in quiescent NSCs and displays a cortical localization. However, this cortical localization is not apparent in Figure 5D, and the authors' claim that Hig "accumulates along the NSC projection" in qNSCs is not fully convincing, as the Hig-positive puncta appear to be randomly distributed. Could the authors clarify why Hig subcellular localization appears to differ between Figures 5C and 5D?

We have provided a clearer lateral view image showing Hig along the qNSC membrane and basal projection.

Quiescent NSCs

Referee #2:

The authors have addressed all of my previous concerns through additional analyses, clarifications, and improved presentation of the data. I am satisfied with the responses and recommend acceptance of the manuscript.

Dear Dr Brand,

Thank you for submitting the revised version of your manuscript. I have now evaluated your amended manuscript and concluded that the remaining minor concerns have been sufficiently addressed.

I am thus pleased to inform you that your manuscript has been accepted for publication in the EMBO Journal.

Best regards,

Daniel Klimmeck

Daniel Klimmeck, PhD
Senior Editor
The EMBO Journal
EMBO
Postfach 1022-40
Meyerhofstrasse 1
D-69117 Heidelberg
contact@embojournal.org

Please note that it is The EMBO Journal policy for the transcript of the editorial process (containing referee reports and your response letters) to be published as an online supplement to each paper. If you should prefer removal of any referee-only figures included in the point-by-point response(s), e.g. because they may still be used for future publication or because they have been reproduced from published work by others, please do let us know immediately via response email.